# Safety and immunogenicity of booster vaccination and fractional dosing with Ad26.COV2.S or BNT162b2 in Ad26.COV2.S-vaccinated participants

Catherine Riou[1,2☯], Jinal N. Bhiman[3,4☯], Yashica Ganga[5☯], Shobna Sawry[6☯], Frances Ayres[3,4], Richard Baguma[1], Sashkia R. Balla[3,4], Ntombi Benede[1], Mallory Bernstein[5], Asiphe S. Besethi[1], Sandile Cele[5], Carol Crowther[3,4], Mrinmayee Dhar[6], Sohair Geyer[1], Katherine Gill[7], Alba Grifoni[8], Tandile Hermanus[3,4], Haajira Kaldine[3,4], Roanne S. Keeton[1], Prudence Kgagudi[3,4], Khadija Khan[5,9], Erica Lazarus[10], Jean Le Roux[6], Gila Lustig[11], Mashudu Madzivhandila[3], Siyabulela F. J. Magugu[1], Zanele Makhado[3,4], Nelia P. Manamela[3,4], Qiniso Mkhize[3,4], Paballo Mosala[1], Thopisang P. Motlou[3,4], Hygon Mutavhatsindi[1], Nonkululeko B. Mzindle[3], Anusha Nana[10], Rofhiwa Nesamari[1], Amkele Ngomti[1], Anathi A. Nkayi[1], Thandeka P. Nkosi[7], Millicent A. Omondi[1], Ravindre Panchia[10], Faeezah Patel[6], Alessandro Sette[8,12], Upasna Singh[11], Strauss van Graan[3,4], Elizabeth M. Venter[3,4], Avril Walters[1], Thandeka Moyo-Gwete[3,4], Simone I. Richardson[3,4], Nigel Garrett[11,13], Helen Rees[6], Linda-Gail Bekker[7], Glenda Gray[14], Wendy A. Burgers[1,2☯], Alex Sigal[5,9,11☯], Penny L. Moore[3,4,11☯], Lee Fairlie[6☯]*

1 Division of Medical Virology, Department of Pathology, Institute of Infectious Disease and Molecular Medicine, University of Cape Town, Cape Town, South Africa, 2 Wellcome Centre for Infectious Diseases Research in Africa, University of Cape Town, Cape Town, South Africa, 3 SA MRC Antibody Immunity Research Unit, School of Pathology, University of the Witwatersrand, Johannesburg, South Africa, 4 Center for HIV and STIs, National Institute for Communicable Diseases of the National Health Laboratory Services, Johannesburg, South Africa, 5 Africa Health Research Institute, Durban, South Africa, 6 Wits RHI, Faculty of Health Sciences, University of the Witwatersrand, Johannesburg, South Africa, 7 The Desmond Tutu HIV Centre, University of Cape Town, Cape Town, South Africa, 8 Center for Vaccine Innovation, La Jolla Institute for Immunology, La Jolla, California, United States of America, 9 School of Laboratory Medicine and Medical Sciences, University of KwaZulu-Natal, Durban, South Africa, 10 Perinatal HIV Research Unit, Faculty of Health Science, University of the Witwatersrand, Johannesburg, South Africa, 11 Centre for the AIDS Programme of Research in South Africa, University of KwaZulu-Natal, Durban, South Africa, 12 Division of Infectious Diseases and Global Public Health, Department of Medicine, University of California, San Diego (UCSD), La Jolla, California, United States of America, 13 Department of Public Health Medicine, School of Nursing and Public Health, University of KwaZulu-Natal, Durban, South Africa, 14 South African Medical Research Council, Cape Town, South Africa

☯ These authors contributed equally to this work.
* LFairlie@wrhi.ac.za

**Data Availability Statement:** All data underlying the findings described in this manuscript are freely

## Abstract

We report the safety and immunogenicity of fractional and full dose Ad26.COV2.S and BNT162b2 in an open label phase 2 trial of participants previously vaccinated with a single dose of Ad26.COV2.S, with 91.4% showing evidence of previous SARS-CoV-2 infection. A total of 286 adults (with or without HIV) were enrolled >4 months after an Ad26.COV2.S prime and randomized 1:1:1:1 to receive either a full or half-dose booster of Ad26.COV2.S or BNT162b2 vaccine. B cell responses (binding, neutralization and antibody dependent

available to other researchers. They have included in the supplementary material as S1 Table.

**Funding:** This study was funded by the South African Medical research Council (SAMRC to LF), the Bill and Melinda Gates Foundation (INV-030570 to PLM, through the Global Immunology and Immune Sequencing for Epidemic Response (GIISER) program and INV-046743 to FL) and the Wellcome Trust (226137/Z/22/Z to ASi). PLM is supported by the SAMRC (96833) and is an Department of Science and Innovation-National Research Foundation South African Research Chair (98341). ASi is supported by the Bill and Melinda Gates foundation (INV-046743) and the SAMRC (D2112300-01). WAB is supported by the EDCTP2 program of the European Union's Horizon 2020 programme (TMA2016SF-1535-CaTCH-22) and the EU-Africa Concerted Action on SARS-CoV-2 Virus Variant and Immunological Surveillance (COVICIS), funded through the EU's Horizon Europe Research and Innovation Programme (101046041). CR is supported by the EDCTP2 program (TMA2017SF-1951-TB-SPEC). This project has also been funded in part by the National Institute of Allergy and Infectious Diseases, NIH, Department of Health and Human Services, under Contract No. 75N93021C00016 to AG and 75N93019C00065 to ASe. The Wits RHI site received grant funding from Janssen to conduct the following clinical trials: Ensemble study (3UM1 AI068614-14SI to FP), the Sisonke 1 study (96833 to FP), the Sherpa Study (96867 to FP) as well as Pfizer for the Pfizer C4591015 study (C4591015 to FP), Horizon 1 (VAC31518COV2004 to LF) and Horizon 2 (VAC31518COV3006 to FL). The funders had no role in study design, data collection and analysis, decision to publish, or preparation of the manuscript. For the purposes of open access, the authors have applied a CC-BY public copyright license to any author-accepted version.

**Competing interests:** I have read the journal's policy and the authors of this manuscript have the following competing interests: A.Se. is a consultant for AstraZeneca Pharmaceuticals, Calyptus Pharmaceuticals, Inc, Darwin Health, EmerVax, EUROIMMUN, F. Hoffman-La Roche Ltd, Fortress Biotech, Gilead Sciences, Granite bio., Gritstone Oncology, Guggenheim Securities, Moderna, Pfizer, RiverVest Venture Partners, and Turnstone Biologics. A.G. is a consultant for Pfizer. LJI has filed for patent protection for various aspects of T cell epitope and vaccine design work. All other authors declare no competing interests.

cellular cytotoxicity-ADCC), and spike-specific T-cell responses were evaluated at baseline, 2, 12 and 24 weeks post-boost. Antibody and T-cell immunity targeting the Ad26 vector was also evaluated. No vaccine-associated serious adverse events were recorded. The full- and half-dose BNT162b2 boosted anti-SARS-CoV-2 binding antibody levels (3.9- and 4.5-fold, respectively) and neutralizing antibody levels (4.4- and 10-fold). Binding and neutralizing antibodies following half-dose Ad26.COV2.S were not significantly boosted. Full-dose Ad26.COV2.S did not boost binding antibodies but slightly enhanced neutralizing antibodies (2.1-fold). ADCC was marginally increased only after a full-dose BNT162b2. T-cell responses followed a similar pattern to neutralizing antibodies. Six months post-boost, antibody and T-cell responses had waned to baseline levels. While we detected strong anti-vector immunity, there was no correlation between anti-vector immunity in Ad26.COV2.S recipients and spike-specific neutralizing antibody or T-cell responses post-Ad26.COV2.S boosting. Overall, in the context of hybrid immunity, boosting with heterologous full- or half-dose BNT162b2 mRNA vaccine demonstrated superior immunogenicity 2 weeks post-vaccination compared to homologous Ad26.COV2.S, though rapid waning occurred by 12 weeks post-boost.

**Trial Registration:** The study has been registered to the South African National Clinical Trial Registry (SANCTR): DOH-27-012022-7841. The approval letter from SANCTR has been provided in the up-loaded documents.

## Introduction

The development of vaccines against SARS-CoV-2 was unparalleled, with numerous platforms suited to rapid production dominating use in initial vaccination programs globally. This included adenovirus-vectored vaccines, such as the Janssen Ad26.COV2.S, replication-incompetent adenovirus 26 vectored SARS-CoV-2 spike protein vaccine. This vaccine is registered as a single dose, with subsequent boosters recommended. This was the first vaccine available nationally in South Africa, and health care workers, and later other essential workers, were offered this vaccine as part of the Sisonke trial [1]. The mRNA-based Pfizer BNT162b2 Comirnaty vaccine became available in South Africa subsequently as part of the national vaccine rollout.

The emergence of SARS-CoV-2 variants of concern (VOCs) including the Beta, Delta and Omicron variants reduced vaccine effectiveness against infection [2–6]. All vaccines based on the sequence of the SARS-CoV-2 ancestral spike, including the Janssen Ad26.COV2.S and the Pfizer Comirnaty BNT162b2 vaccines, elicited dramatically lower titers of neutralizing antibodies against the Omicron subvariants [4, 7–20]. Vaccines incorporating Omicron subvariant sequences [21] are not available in South Africa. However, the main driver of increased neutralizing capacity against Omicron variants is hybrid immunity, which is the combination of vaccine and infection elicited immunity. Population studies in SA showed seroprevalence levels in excess of 95% by the end of the Omicron BA.1 Wave [22].

However, given the waning of humoral and cellular immunity over time, COVID-19 vaccine boosting may be beneficial, especially in individuals at high risk for severe disease. The choice of booster, timing and dose remain largely dependent on regulatory and national considerations, including fiscal constraints and capacity of the health care system. Several studies have demonstrated a more robust humoral and cellular immune response with a heterologous

boost compared to a homologous boost, in particular when boosting is with an mRNA vaccine [12, 15, 23]. A further consideration is that due to pressure on vaccine development, cost and equitable access, which are significantly impacted if multiple boosters are needed, strategies such as fractional dosing should be considered. Fractional dosing has previously been used with other vaccines such as yellow fever [24].

South Africa is burdened by more HIV infections than any other country in the world, with approximately 8 million people living with HIV (PLWH) [25, 26]. PLWH, especially those with low CD4 T-cell counts, have moderately worse COVID-19 outcomes [27–35]. This is associated with lower and delayed neutralizing antibody titers in response to SARS-CoV-2 infection [19, 34, 36], more pronounced in PLWH with HIV viremia [19]. However, similar to results reported for the AstraZeneca ChAdOx and the Pfizer BNT162b2 vaccines [37–41], there was no observed difference between PLWH and HIV-negative individuals in neutralizing antibody titers against SARS-CoV-2 spike after Ad26.COV2.S vaccination [19, 36].

Vaccines which use a viral vector to deliver the immunogen may be inhibited by pre-existing immunity to the virus on which the vector is based [42, 43]. In addition to eliciting immunity to the vaccine target, vaccination with an adenovirus vectored vaccine has been shown to elicit neutralizing antibody and T-cell immunity to the vector itself [44–46]. This may reduce the ability of repeated doses of the vectored vaccine to infect cells or reduce vaccine vector-infected cell survival. The degree of elicited anti-vector immunity could therefore potentially determine the effectiveness of vectored immunization in a population with previous immunity to the virus on which the vector is based, or effectiveness if the vaccinees have been previously immunized with the same vector. However additional factors, including transgene persistence as well as local or systemic persistence may also impact the effectiveness of viral vectored vaccines.

In this study, we evaluated the immunogenicity and safety of diverse boost strategies after primary Ad26.COV2.S vaccination. We tested fractional and full dose heterologous and homologous booster Ad26.COV2.S and BNT162b2 vaccinations. We also determined the effects of HIV status, and degree of anti-adenovirus 26 vector immunity before and after boosting. We observed that heterologous boosting of Ad26.COV2.S with the BNT162b2 mRNA vaccine was the most effective regimen to transiently enhance humoral and T-cell immunity and HIV status did not have a substantial effect on the outcome. We show that homologous boosting with Ad26.COV2.S occurred in the context of high anti-Ad26 vector immunity. Although, Ad26.COV2.S gave a weak increase in antibody and cellular responses against SARS-CoV-2 spike, we observed no correlation between vaccine response and anti-Ad26 neutralizing antibody levels, as previously reported [47, 48].

## Methods and materials

### Study design and participant

This study, **B**ooster **A**fter **Si**sonke **S**tudy (BaSiS), is an ongoing phase 2, randomized, open-label trial to evaluate the safety and immunogenicity of a booster vaccination in participants who received a single Ad26.COV2.S vaccine through the Sisonke phase 3B implementation study or via the South African National Department of Health COVID-19 vaccine rollout (S1 Protocol and S1 Checklist). Participants were healthy adults, who met study eligibility criteria and included participants with well-controlled comorbidities, except for HIV infection where there were no immunological or virological exclusions. Full eligibility criteria are provided in the protocol. We aimed to enrol at least one third of participants as PLWH and at least 10% >55 years of age. Participants were included if they had no SARS-CoV-2 infection at least 28 days prior to randomization. The study was conducted at four research sites in South Africa,

two in Johannesburg (Perinatal HIV Research Unit (PHRU), The Wits Reproductive Health and HIV Institute (WITS RHI)), one in Durban (Centre for the AIDS Programme of Research in South Africa (CAPRISA)) and one in Cape Town (Desmond Tutu Health Foundation). Participants were recruited between 15/12/2021 and 30/06/2022 at PHRU; between 08/12/2021 and 30/06/2022 at WITS RHI; between 04/02/2022 and 08/06/2022 at CAPRISA and between 23/03/2022 and 27/07/2022 at the Desmond Tutu Health Foundation.

## Randomization

Participants were randomized 1:1:1:1 to one of four booster vaccinations including: Arm A: full-dose Ad26.COV2.S ($5x10^{10}$ vp/mL, 0.25 mL); Arm B: half-dose Ad26.COV2.S ($2.6x10^{10}$ vp/mL, 0.13 mL); Arm C: full-dose BNT162b2 Comirnaty vaccine (30mcg) and Arm D: half-dose BNT162b2 Comirnaty vaccine (15mcg). No masking was required since the study was open-label.

## Procedures

Study visits took place at baseline (randomization), 2 weeks, 12 weeks and 24 weeks. At each visit, medical history, COVID-19 infection history (symptoms or positive test) and vaccination history (COVID-19 vaccine or other vaccination) was taken, followed by a targeted clinical examination when necessary. At the baseline visit, participants received a single booster vaccination, as per the randomization arm. Diary cards were issued and participants were trained to collect data up to 7 days post booster. Telephonic contact at day 7 was conducted to enquire about severity or ongoing nature of reactogenicity. Blood samples were drawn at each visit and included HIV testing at baseline (in all except those known to be PLWH), CD4 count and HIV viral load (in PLWH), full blood count (BL, W2 and additional visits in PLWH), D-dimers (BL and W2). A nasopharyngeal swab for SARS-CoV-2 PCR was conducted at baseline. Interim visits were held for any safety concerns or if a participant had COVID-19 symptoms or a positive SARS-CoV-2 PCR or antigen test outside of the study.

## Safety

Reactogenicity was collected and graded according to criteria included in Food and Drug Administration (FDA) "The Guidance for Industry: Toxicity Grading Scale for Healthy Adult and Adolescent Volunteers Enrolled in Preventive Vaccine Clinical Trials" [49]. All grade 3 reactogenicity events, SAEs, SUSARs and adverse drug reactions were reported. A Data Safety and Monitoring Committee was established to evaluate protocol-defined safety events and immunogenicity data.

## Immunogenicity

**Live virus neutralization assay.** Live virus neutralization assay was performed as previously described in our previous work [4, 6, 18]. Briefly, ACE2-expressing H1299-E3 (CRL-5803, ATCC) cells were seeded at $4.5×10^5$ cells per well and incubated for 18–20 h. After washing, the sub-confluent cell monolayer was inoculated with 500 μL universal transport medium diluted 1:1 with filtered growth medium. Cells were incubated for 1 h. Wells were then filled with 3 mL complete growth medium. After 4 days of infection (completion of passage 1), cells were trypsinized, centrifuged and resuspended in 4 mL growth medium. Then, all infected cells were added to Vero E6 cells (CRL-1586, ATCC). The coculture of ACE2-expressing H1299-E3 and Vero E6 cells was incubated for 4 days. The viral supernatant from this culture (passage 2 stock) was used for experiments.

H1299-E3 cells were plated at 30,000 cells/well 1 day pre-infection. Aliquots of cryo-preserved plasma samples were heat-inactivated and clarified by centrifugation. Virus stocks were used at approximately 50–100 focus-forming units per microwell and added to diluted plasma. Antibody–virus mixtures were incubated for 1 h at 37˚C. Cells were infected with 100 µL of the virus–antibody mixtures for 1 h, then 100 µL of a RPMI 1640 (Sigma-Aldrich), 1.5% carboxymethylcellulose (Sigma-Aldrich) overlay was added without removing the inoculum. Cells were fixed 18 h post-infection. Foci were stained with a rabbit anti-spike monoclonal antibody (0.5 µg/mL, BS-R2B12, GenScript) overnight at 4˚C, washed and then incubated with a horseradish peroxidase (HRP) conjugated goat anti-rabbit antibody (1 µg/mL, Abcam ab205718) for 2 h. TrueBlue peroxidase substrate (SeraCare) was then added and incubated for 20 min. Plates were imaged in an ImmunoSpot Ultra-V S6-02-6140 Analyzer ELISPOT instrument with BioSpot Professional built-in image analysis (C.T.L). All statistics and fitting were performed using custom code in MATLAB v.2019b. Neutralization data were fit to: $Tx = 1/1+(D/ID_{50})$. Tx is the number of foci normalized to the number of foci in the absence of plasma on the same plate at dilution D and $ID_{50}$ is the plasma dilution giving 50% neutralization. $FRNT_{50} = 1/ID_{50}$. Values of $FRNT_{50} < 1$ are set to 1 (undiluted), the lowest measurable value. As, the most concentrated plasma dilution was 1:25, $FRNT_{50} < 25$ were extrapolated. To calculate confidence intervals, $FRNT_{50}$ or fold-change in $FRNT_{50}$ per participant was log-transformed and arithmetic mean plus and minus two standard deviations were calculated for the log transformed values. These were exponentiated to obtain the upper and lower 95% confidence intervals on the geometric mean $FRNT_{50}$ or the fold-change in $FRNT_{50}$ geometric means.

Sequences of outgrown ancestral SARS-CoV-2 and the Omicron BA.5 subvariant have been deposited in GISAID with accession EPI_ISL_602626.1 (ancestral, D614G) and EPI_ISL_12268493.2 (Omicron/BA.5).

**SARS-CoV-2 spike and nucleocapsid enzyme-linked immunosorbent assay (ELISA).**
For ELISA, Hexapro SARS-CoV-2 full spike protein with the D614G substitution were expressed in Human Embryonic Kidney (HEK) 293F suspension cells by transfecting the cells with the respective expression plasmid. After incubating for 6 days at 37˚C, proteins were first purified using a nickel resin followed by size exclusion chromatography. Relevant fractions were collected and frozen at -80˚C until use. Two µg/mL of D614G spike or nucleocapsid protein was used to coat 96-well, high-binding plates (Corning) and incubated overnight at 4˚C. The plates were incubated in a blocking buffer consisting of 1x PBS, 5% skimmed milk powder, 0.05% Tween 20. Plasma samples were diluted to 1:100 starting dilution in a blocking buffer and added to the plates. IgG secondary antibody (Merck) was diluted to 1:3000 in blocking buffer and added to the plates followed by TMB substrate (Thermofisher Scientific). Upon stopping the reaction with 1 M H2SO4, absorbance was measured at 450 nm. For spike ELISA, mAbs CR3022 was used as a positive control and Palivizumab was used as a negative control.

**Lentiviral pseudovirus production and neutralization assay.** Virus production and pseudovirus neutralization assays were done as previously described. Briefly, 293T/ACE2.MF cells modified to overexpress human ACE2 (provided by M. Farzan, Scripps Research) were cultured in DMEM (Gibco) containing 10% FBS and 3 µg/mL of puromycin at 37˚C. Cell monolayers were disrupted at confluency by treatment with 0.25% trypsin in 1 mM EDTA (Gibco). The SARS-CoV-2, Wuhan-1 spike, cloned into pCDNA3.1 was mutated using the QuikChange Lightning Site-Directed Mutagenesis kit (Agilent Technologies) and NEBuilder HiFi DNA Assembly Master Mix (NEB) to include D614G (wild-type) or lineage defining mutations for Beta (L18F, D80A, D215G, 241-243del, K417N, E484K, N501Y, D614G and A701V), Delta (T19R, 156-157del, R158G, L452R, T478K, D614G, P681R and D950N), Omicron BA.1 (A67V, Δ69–70, T95I, G142D/Δ143–145, Δ211/L212I, ins214EPE, G339D, S371L,

S373P, S375F, K417N, N440K, G446S, S477N, T478K, E484A, Q493R, G496S, Q498R, N501Y, Y505H, T547K, D614G, H655Y, N679K, P681H, N764K, D796Y, N856K, Q954H, N969K, L981F) and Omicron BA.4/5 (T19I, L24S, 25-27del, 69-70del,G142D, V213G, G339D, S371F, S373P, S375F, T376A, D405N, R408S, K417N,N440K, L452R, S477N, T478K, E484A,F486V, Q498R, N501Y, Y505H, D614G, H655Y, N679K, P681H, N764K, D796Y, Q954H, N969K). Pseudoviruses were produced by co-transfection in 293T/17 cells with a lentiviral backbone (HIV-1 pNL4.luc encoding the firefly luciferase gene) and either of the SARS-CoV-2 spike plasmids with PEIMAX (Polysciences). Culture supernatants were clarified of cells by a 0.45 μM filter and stored at -70˚C. Plasma samples were heat-inactivated and clarified by centrifugation. Pseudovirus and serially diluted plasma/sera were incubated for 1 h at 37˚C. Cells were added at $1 \times 10^4$ cells per well after 72 h of incubation at 37˚C. Luminescence was measured using PerkinElmer Life Sciences Model Victor X luminometer. Neutralization was measured as described by a reduction in luciferase gene expression after single-round infection of 293T/ACE2.MF cells with spike-pseudotyped viruses. Titers were calculated as the reciprocal plasma dilution ($ID_{50}$) causing 50% reduction of relative light units.

**Antibody-dependent cellular cytotoxicity (ADCC) assay.** The ability of plasma antibodies to cross-link FcγRIIIa (CD16) and spike expressing cells was measured as a proxy for ADCC as previously described. HEK293T cells were transfected with 5 μg of SARS-CoV-2 wild-type variant spike (D614G), Beta, Delta and Omicron BA.1 spike plasmids using PEI-MAX 40,000 (Polysciences) and incubated for 2 days at 37˚C. Expression of spike was confirmed by binding of CR3022 and P2B-2F6 and their detection by anti-IgG APC (Biolegend) measured by flow cytometry. Subsequently, $1 \times 10^5$ spike-transfected cells per well were incubated with heat inactivated plasma (1:100 final dilution) or control mAbs (final concentration of 100 μg/mL) in RPMI 1640 media supplemented with 10% FBS and 1% Pen/Strep (R10; Gibco) for 1 h at 37˚C. Jurkat-Lucia NFAT-CD16 cells (Invivogen) ($2 \times 10^5$ cells/well) were added and incubated for 24 h at 37˚C. Twenty μl of supernatant was then transferred to a white 96-well plate with 50 μl of reconstituted QUANTI-Luc secreted luciferase and read immediately on a Victor 3 luminometer with 1s integration time. Cells were gated on singlets, live cells (determined by Live/dead Viability dye; Thermofisher Scientific), and those cells that were positive for IgG and spike specific monoclonal antibodies binding to their surface. Relative light units (RLU) of a no antibody control were subtracted as background. Palivizumab was used as a negative control, while CR3022 was used as a positive control, and P2B-2F6 to differentiate the Beta from the D614G variant. To induce the transgene, 1x cell stimulation cocktail (Thermofisher Scientific) and 2 μg/ml ionomycin in R10 was added as a positive control.

**Measurement of antigen-specific T cells by flow cytometry.** T-cell responses to SARS-CoV-2 spike or human adenovirus 26 (Ad26) hexon and penton were measured as previously described [3]. Briefly, cryopreserved PBMC were thawed, washed and rested in RPMI 1640 (Sigma-Aldrich) containing 10% heat-inactivated FBS (HyClone) for 4 h prior to stimulation. PBMC were seeded in a 96-well V-bottom plate at ~$2 \times 10^6$ PBMC per well and stimulated with either a commercial ancestral SARS-CoV-2 spike (S) pool (1 μg/mL, Miltenyi Biotec) or a Ad26 peptide pool containing 293 peptides (15mers with 10-aa overlap) spanning the Ad26 hexon and penton proteins (1 μg/mL). All stimulations were performed in the presence of Brefeldin A (10 μg/mL, Sigma-Aldrich) and co-stimulatory antibodies against CD28 (clone 28.2) and CD49d (clone L25) (1 μg/mL each, BD Biosciences). As a negative control, PBMC were incubated with co-stimulatory antibodies, Brefeldin A and an equimolar amount of DMSO. After 16 h of stimulation, cells were washed, stained with LIVE/DEAD Fixable Near-IR Stain (Invitrogen) and subsequently surface stained with the following antibodies: CD14 APC-Cy7 (HCD14), CD19 APC-Cy7 (HIB19), CD4 BV785 (OKT4), CD8 FITC (RPA-T8), CD45RA

BV570 (HI100) (Biolegend) and CD27 PE-Cy5 (1A4, Beckman Coulter). Cells were then fixed and permeabilized using a Cytofix/Cytoperm buffer (BD Biosciences) and stained with CD3 BV650 (OKT3), IFN-g BV711 (4S.B3), TNF-a PE-Cy7 (Mab11) and IL-2 PE/Dazzle 594 (MQ1-17H12) from Biolegend. Finally, cells were washed and fixed in CellFIX (BD Biosciences). Samples were acquired on a BD Fortessa flow cytometer and analyzed using FlowJo (v10.8.1, FlowJo LLC). Results are expressed as the frequency of total memory CD4+ or CD8+ T cells expressing IFN-g, TNF-a or IL-2. Due to high TNF-a backgrounds, cells producing TNF-a alone were excluded from the analysis. All data are presented after background subtraction.

**Outcomes.** The primary objectives of the study were to evaluate safety and reactogenicity and humoral and cellular immunogenicity to full and half dose homologous and heterologous booster vaccinations, at each study visit. Primary endpoint measures included measuring nucleocapsid binding antibody titers, neutralization titers and T-cell response magnitudes. Safety was measured by participant self-report using diary cards and graded according to FDA standards [49]. The primary immunogenicity endpoint was defined as any study arm eliciting <75% of the highest geometric mean titre (GMT) response in the study. Together with the DSMB, the study team used this criteria to determine whether a subsequent booster should be offered to participants, and which booster if so. In this study, based on results, all participants except those who received the full dose BNT162b2 vaccination, were offered a full dose BNT162b2 booster in addition to that received according to their original randomisation.

## Statistical analysis

A two-tailed Wilcoxon signed-rank test or a Friedman test with Dunn's correction was used to assess statistical differences between paired samples. A Mann-Whitney test or a Kruskal-Wallis test with Dunn's corrections was used to compare multiple groups. Correlations were tested by a two-tailed non-parametric Spearman's rank test. In all cases, P values of less than or equal to 0.05 were considered significant.

## Data sharing

Protocol may be obtained from the lead authors upon request. All data underlying the findings described in this manuscript are available in S1 Table.

## Study approval

This study has been approved by the South African Health Products Regulatory Authority (SAHPRA, number: 20210423) and all site-specific Human Research Ethics Committees (Wits: 211001B, UKZN: BREC/00003487/2021, UCT: 680/202). All participants provided written informed consent.

## Results

### Participants and follow up

Between December 8, 2021, and July 27, 2022, 333 participants were screened and 289 randomized to 4 arms: Arm A: Full-dose Ad26.COV2.S ($5x10^{10}$ vp/mL, 74 participants); Arm B: half-dose Ad26.COV2.S ($2.6x10^{10}$ vp/mL, 69 participants); Arm C: Full-dose BNT162b2 (30mcg, 73 participants); and Arm D: half-dose BNT162b2 (15mcg, 73 participants) (Table 1 and Fig 1). Participants were followed for 24 weeks, overall retention was 93.1% at 24 weeks (Table 1).

**Table 1. Clinical characteristics of study participants.**

| | Arm A | Arm B | Arm C | Arm D | Total |
|---|---|---|---|---|---|
| | Full-dose Ad26.COV2.S | Half-dose Ad26.COV2.S | Full-dose BNT162b2 | Half-dose BNT162b2 | |
| Vaccinated (N) | 74 | 69 | 73 | 73 | 289 |
| **Age\*** | 42 (35–48) | 40 (35–45) | 43 (35–50) | 42 (35–49) | 42 (35–49) |
| 18–29 years | 5 (6.8%) | 5 (7.2%) | 7 (9.6%) | 6 (8.2%) | 23 (8%) |
| 30–44 years | 41 (55.4%) | 46 (66.7%) | 34 (46.6%) | 34 (46.6%) | 155 (53.6%) |
| 45–54 years | 20 (27%) | 10 (14.5%) | 23 (31.5%) | 22 (30.1%) | 75 (26%) |
| $\geq$ 55 years | 8 (10.8%) | 8 (11.6%) | 9 (12.3%) | 11 (15.1%) | 36 (12.5%) |
| **Sex** | | | | | |
| Male | 10 (13.5%) | 8 (11.6%) | 18 (24.7%) | 15 (20.5%) | 51 (17.6%) |
| Female | 64 (86.5%) | 61 (88.4%) | 55 (75.3%) | 58 (79.5%) | 238 (82.4%) |
| **Ethnicity** | | | | | |
| Black African | 69 (93.2%) | 67 (97.1%) | 69 (94.5%) | 70 (95.9%) | 275 (95.2%) |
| Other | 5 (6.8%) | 2 (2.9%) | 4 (5.5%) | 3 (4.1%) | 14 (4.8%) |
| Days between prime and Booster\* | 265 (245–292) | 271 (259–302) | 273 (258–291) | 272 (253–300) | 271 (255–296) |
| **BMI** | | | | | |
| Underweight | 0 (0%) | 2 (2.9%) | 0 (0%) | 1 (1.4%) | 3 (1%) |
| Normal | 12 (16.2%) | 8 (11.6%) | 14 (19.2%) | 6 (8.2%) | 40 (13.8%) |
| Overweight | 20 (27%) | 41 (59.4%) | 39 (53.4%) | 47 (64.4%) | 71 (24.6%) |
| Obese | 41 (55.4%) | 39 (56.5%) | 47 (64.4%) | 47 (64.4%) | 174 (60.2%) |
| Not available | 1 (1.4%) | 0 (0%) | 0 (0%) | 0 (0%) | 0 (0%) |
| **HIV infection** | | | | | |
| HIV Positive | 27 (36.5%) | 28 (40.6%) | 31 (42.5%) | 30 (41.1%) | 116 (40.1%) |
| Viremic$ | 1 (3.7%) | 5 (17.9%) | 4 (12.9%) | 6 (20%) | 16 (13.8%) |
| CD4 count (cells/mm$^3$)\* | 223 (na) | 452 (103–471) | 1136 | 341 (132–538) | 452 (132–538) |
| Viral load (copies/mL)\* | 818 (na) | 5838 (3054–18252) | 29911 (14863–47207) | 19960 (5324–37713) | 15376 (3054–27420) |
| Aviremic$ | 26 (96.3%) | 23 (82.1%) | 27 (87.1%) | 24 (80%) | 100 (86.2%) |
| CD4 count (cells/mm$^3$)\* | 698 (580–923) | 677 (530–855) | 812 (623–931) | 666 (538–737) | 708 (558–922) |
| **Prior SARS-CoV-2 infection[&]** | 51/58 (87.8%) | 52/57 (91.2%) | 55/59 (93.2%) | 55/59 (93.2%) | 213/223 (91.4%) |
| **Co-morbidities** | | | | | |
| Hypertension | 19 (25.7%) | 8 (11.6%) | 17 (23.3%) | 17 (23.3%) | 61 (21.1%) |
| Anaemia | 7 (9.5%) | 8 (11.6%) | 6 (8.2%) | 9 (12.3%) | 30 (10.4%) |
| Asthma | 6 (8.1%) | 3 (4.3%) | 2 (2.7%) | 1 (1.4%) | 12 (4.2%) |
| Diabetes mellitus | 3 (4.1%) | 1 (1.4%) | 4 (5.5%) | 4 (5.5%) | 12 (4.2%) |
| Arthritis | 1 (1.4%) | 0 (0%) | 3 (4.1%) | 3 (4.1%) | 7 (2.4%) |
| Tuberculosis | 0 (0%) | 4 (5.8%) | 2 (2.7%) | 0 (0%) | 6 (2.1%) |
| **Retention** | | | | | |
| W2 visits completed | 74 (100%) | 66 (95.7%) | 72 (98.6%) | 69 (94.5%) | 281 (97.2%) |
| W12 visits completed | 74 (100%) | 67 (97.1%) | 67 (91.8%) | 70 (95.9%) | 278 (96.2%) |
| W24 visits completed | 69 (93.2%) | 63 (91.3%) | 69 (94.5%) | 68 (93.2%) | 269 (93.1%) |

Unless specified, all data are presented as n (%N);

\*median and interquartile range (IQR);

$% is of all HIV-infected participants; Viremic individuals were defined as having a HIV-1 viral load (VL) >200 HIV mRNA copies/mm$^3$; 3 participants in the viremic arm C had a missing CD4 count;

[&]Prior SARS-CoV-2 infection was defined by the presence of Nucleocapsid-specific IgG; na: non-applicable.

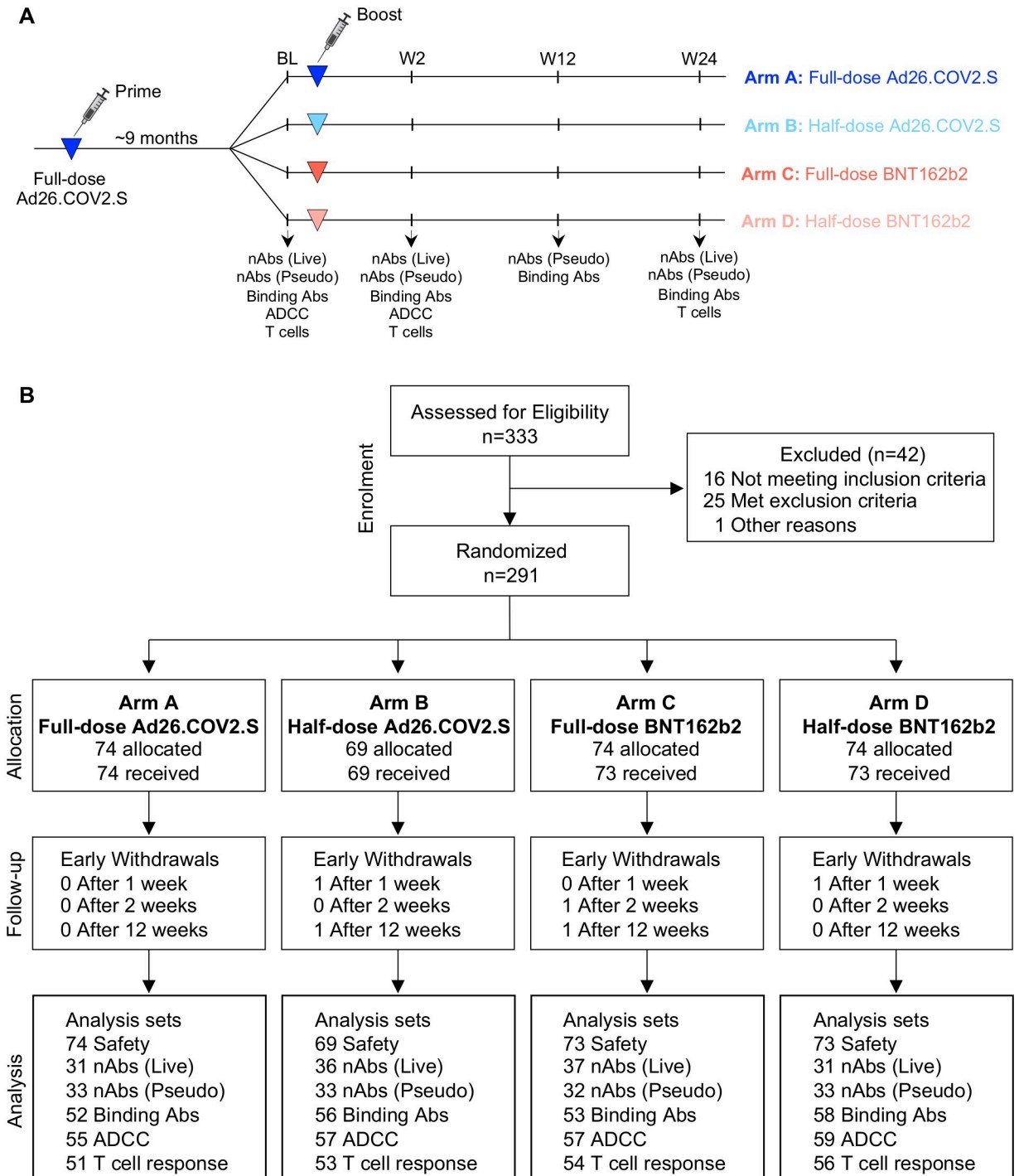

**Fig 1. Study design and CONSORT diagram. (A)** Study design. **(B)** CONSORT flow diagram. BL: baseline, nAbs (Live): Live virus neutralization assay; nAbs (Pseudo): pseudovirus neutralization assay. Binding Abs: Spike-specific IgG ELISA. ADCC: Antibody-dependent cellular cytotoxicity assay. T-cell response: Spike-specific T-cell intracellular cytokine staining assay.

The majority of participants (53.6%, 155/289) were between 30 and 45 years old, with 36/189 (12.5%) 55 years and older, and 238/289 (82.4%) female. In the cohort, 116/289 (40.1%) were PLWH, 16 (13.8%) were considered viremic (VL >200cps/ml) and had a median CD4 count of 452 cells/mm$^3$ (IQR: 132–538), whereas participants who were virologically suppressed (VL <200cps/ml) had a median CD4 count of 708 cells/mm$^3$ (IQR: 558–922). At randomisation, subsequently referred to as baseline, 61/289 (21.1%) of participants had hypertension, 30/289 (10.4%) anaemia, and 174/289 (60.2%) were obese. Other comorbidities were less common (asthma and diabetes mellitus 4.2% each, arthritis 2.4% and tuberculosis 2.1% respectively).

Evidence of previous SARS-CoV-2 infection (positive nucleocapsid antibody detected by ELISA) was present in 91.4% (213/223) of participants tested at baseline indicating a high degree of hybrid immunity in the trial cohort. This was similar amongst the four vaccination arms, namely 91.2% (52/57), 87.9% (51/58), 93.2% (55/59) and 93.2% (55/59) for half-dose Ad26.COV2.S, full-dose Ad26.COV2.S, half-dose BNT162b2 and full-dose BNT162b2, respectively. These high levels of nucleocapsid positivity were sustained 2, 12 and 24 weeks after the booster dose, with greater than 80% nucleocapsid positivity at any given time point. Participants received the booster vaccine at a median of 271 days (IQR: 255–296) after the Ad26.COV2.S prime. Nine breakthrough infections (BTI) were confirmed by SARS-CoV-2 PCR during the trial and all were mild infections that resolved within 4–12 days, 7/9 in the Ad26.COV2.S arms. Three BTI occurred between W2 and W12, and the remainder between W12 and W24 (median: 17.1 weeks post boost).

## Safety and reactogenicity of the booster vaccination regimens

Reactogenicity was measured through participant-completed diary cards, recording solicited local and systemic adverse reactions, as well as unsolicited adverse reactions through day-7 post booster, or longer if adverse events persisted. Overall, safety profiles were comparable between the four trial arms. Localised pain, headache, localised tenderness and weakness were reported with highest frequency, mostly of grade 1 and 2 severity. Grade 3 or 4 events included localised pain (2.0%), tenderness (1.7%), nausea (0.7%), diarrhoea (1.0%), headache (1.7%), weakness (2.8%), myalgia (1.0%), chills (0.7%), cough (0.3%), loss of smell (0.3%) and loss of taste (0.3%). Unsolicited AEs were uncommon, occurring in 8.7% of participants (Fig 2).

No events of thrombosis with thrombocytopenia syndrome (TTS) were reported; thrombocytopenia was reported in three participants, two had thrombocytopaenia at baseline. There were 14 serious AEs (SAEs) on study, non-related to study product, and no AEs of special interest (AESIs).

## Antibody responses after booster vaccination

We evaluated spike binding antibody titers at baseline (BL) and week 2 (W2), week 12 (W12) and week 24 (W24) post-boost in the four vaccination arms using a trimeric spike ELISA for ancestral D614G virus. Relatively high spike binding antibody titers were detected in all groups at BL (geometric mean titer (GMT), EC$_{50:}$ >1000; Fig 3A). This is consistent with our observation that the majority of participants showed evidence of prior infection (Table 1). Spike antibody titers in the half- and full-dose Ad26.COV2.S arms were not significantly boosted by W2 (1.15 and 1.29 fold change in GMT, respectively), and did not change significantly up to W24. In contrast, half- and full-dose BNT162b2 arms demonstrated a significant increase in binding antibody titers at W2 (3.9 and 4.54 fold change, respectively), to a GMT >4500 (Fig 3A). Spike binding antibodies in the BNT162b2 arms reduced to half that of W2 titers by W12, and had returned to BL levels by W24. These kinetics were reflected in a significantly higher fold-

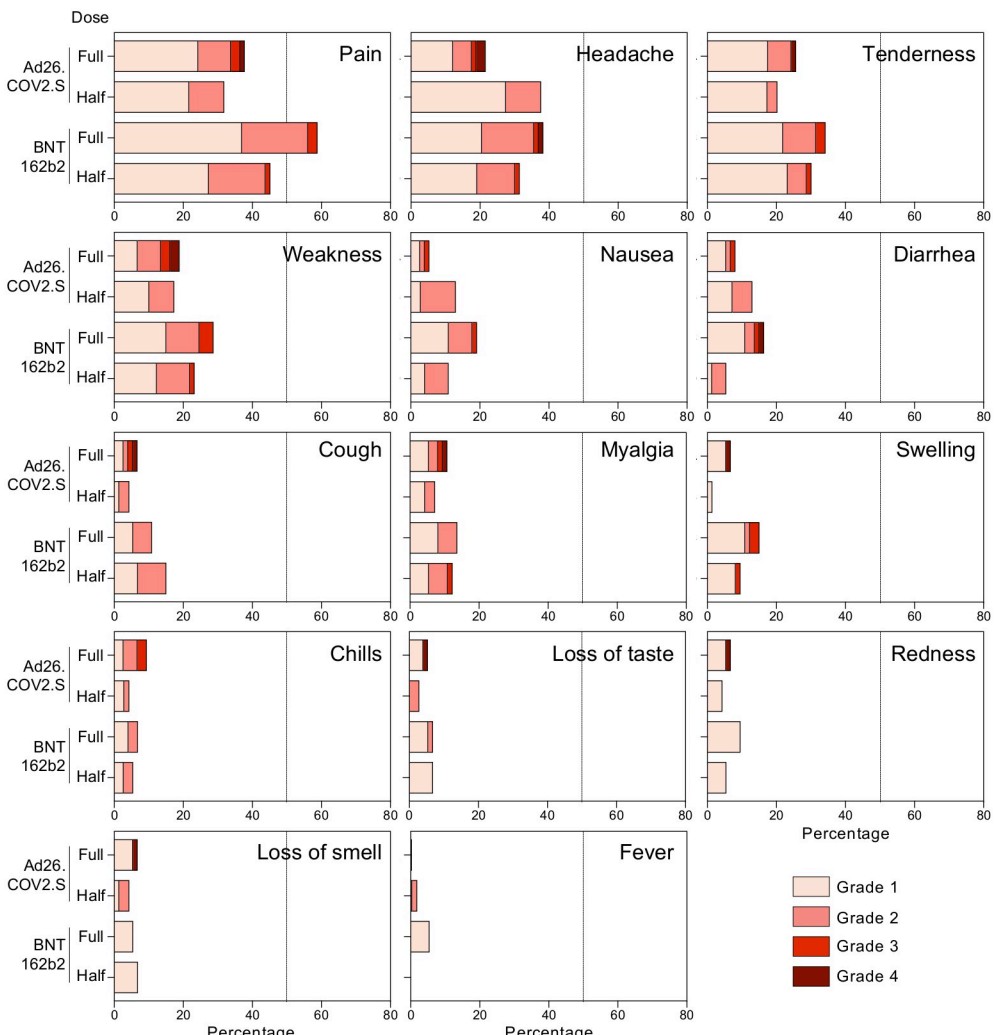

**Fig 2. Recorded adverse events in each study arm.** Distribution of participants experiencing adverse events (pain, headache, tenderness, weakness, nausea, diarrhea, cough, myalgia, swelling, chills, loss of taste, redness, loss of smell or fever) recorded 1 week post booster vaccination in each study arm.

change between W2 and BL in the BNT162b2 arms compared to the Ad26.COV2.S arms (Fig 3B). Although waning had occurred by W12, $EC_{50}$ titers in the BNT162b2 arms were still significantly higher than the Ad26.COV2.S arms (Fig 3C). While some differences between the arms persisted at W24, relatively high antibody titers, similar to those observed at BL and prior to a boost, were noted in all four arms, with GMT of 1016, 1140, 1563 and 1686 for half-dose Ad26.COV2.S, full-dose Ad26.COV2.S, half-dose BNT162b2 and full-dose BNT162b2, respectively.

Using a live virus neutralization assay, we evaluated neutralizing antibody titers to the ancestral D614G virus and Omicron BA.5, which was the most recent dominant sub-lineage at the time of W24 collection. At BL, neutralization activity against the D614G virus was similar between the arms (GMT $FRNT_{50}$ values of 546, 345, 393 and 334 for half-dose Ad26.COV2.S, full-dose Ad26.COV2.S, half-dose BNT162b2 and full-dose BNT162b2, respectively; Fig 4A). When measured at W2, half-dose Ad26.COV2.S did not boost BL neutralizing titers, while

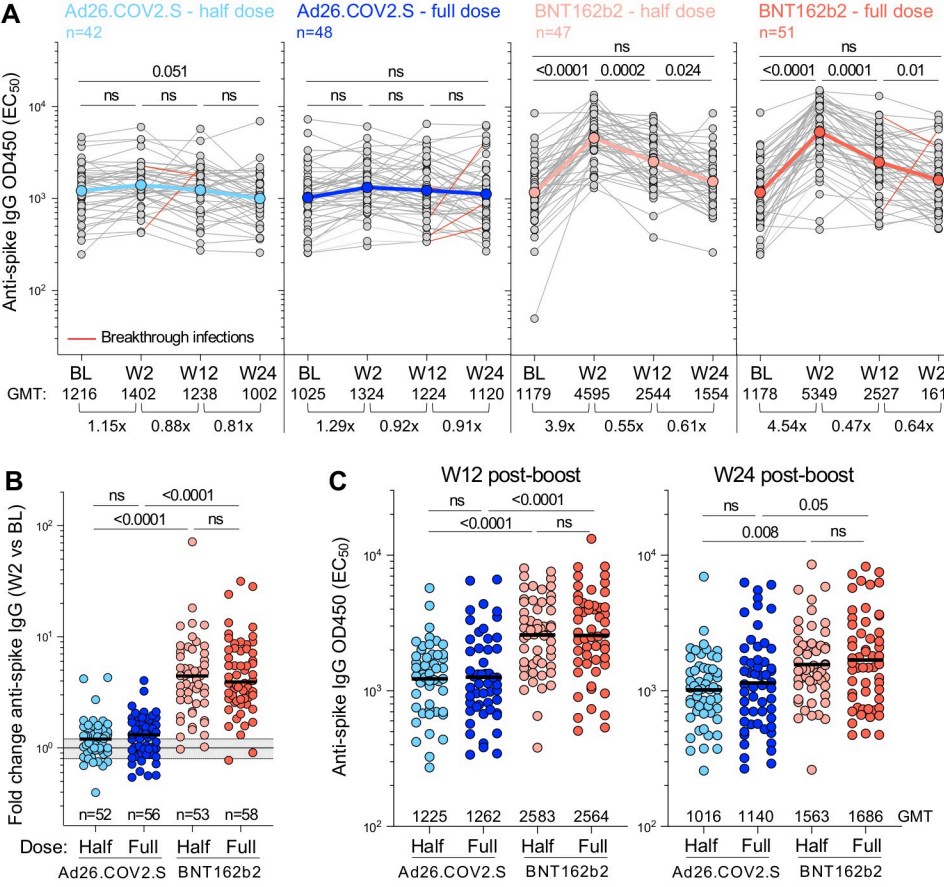

**Fig 3. Spike-specific IgG responses over time in the immunogenicity sub-study population. (A)** Longitudinal spike-specific IgG titer ($EC_{50}$) at baseline (BL), W2, W12 and W24 after vaccine booster. The color-coded dots and bold lines represent the geometric mean titer (GMT) at each time point. Recorded BTI between W2 and W24 are depicted with a red line. Fold-change in the GMT is indicated at the bottom of each graph. Statistical comparisons were performed using a Friedman test with Dunn's correction. **(B)** Fold change in spike-specific IgG titer between W2 and BL in each study arm. Bars represent median fold change. Statistical comparisons were performed using a Kruskal-Wallis test with Dunn's correction. **(C)** Comparison of spike-specific IgG titer between study arms at W12 (left panel) and W24 (right panel). Bars represent GMT. Statistical comparisons (in B and C) were performed using a Kruskal-Wallis test with Dunn's correction.

full-dose Ad26.COV2.S led to a 2.14-fold increase (Fig 4B). Both doses of BNT162b2 demonstrated a superior ability to boost neutralizing antibodies at W2 (8.5 to 10.2-fold compared to BL), with all but one participant increasing neutralizing titer (Fig 4B). As expected from its known immune evasion properties, $FRNT_{50}$ values for BA.5 neutralization were considerably lower than for ancestral virus (GMT of 84, 75, 84 and 63 for the four arms, respectively; Fig 4C). The kinetics mirrored those for D614G neutralization, with both doses of BNT162b2 demonstrating enhanced boosting of neutralizing antibodies compared to Ad26.COV2.S (Fig 4D). By week 24, neutralizing titers had waned significantly for the D614G virus, to a GMT of 732 and 840 for the BNT162b2 half- and full-doses, which was significantly higher than full-dose Ad26.COV2.S, with a GMT of 292 (Fig 4E). Neutralization of BA.5 was poor at W24 (GMT of 73–192), with no differences between the groups. Five participants with documented BTI between W2 and W24 (Fig 4A and 4C, red lines) showed boosted neutralization activity, as expected.

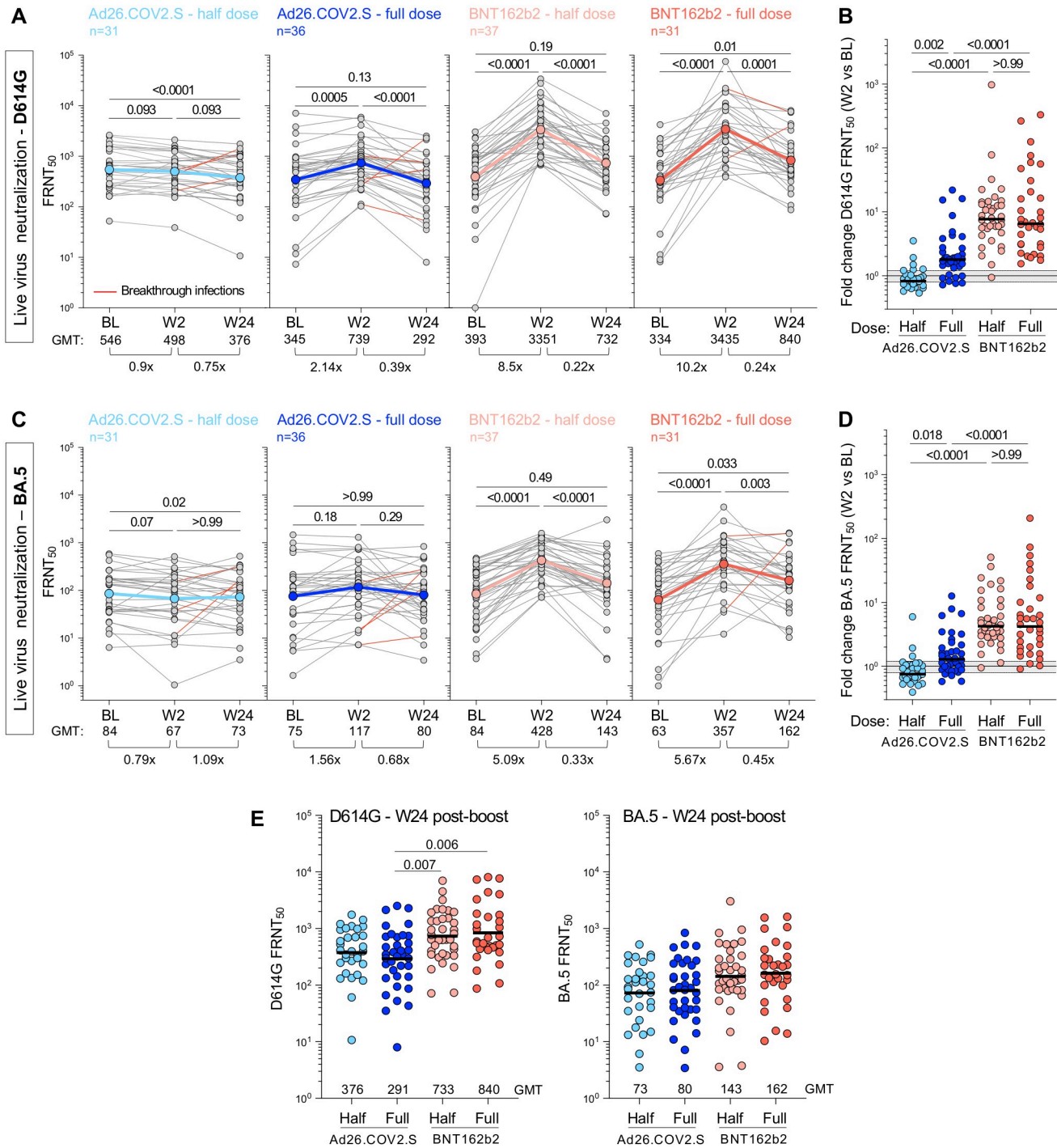

**Fig 4. Live-virus neutralization activity against ancestral D614G and BA.5 SARS-CoV-2 variant after booster vaccination.** Neutralizing titer ($FRNT_{50}$) against ancestral D614G **(A&B)** and Omicron BA.5 **(C&D)** at BL and post-vaccine booster. **A&C** show titer at BL, W2 and W24 post-boost for D614G **(A)** and BA.5 **(C)**. Fold-change of the GMT is indicated at the bottom of each graph. The color-coded dots and bold lines represent the GMT at each time point. Recorded BTI between W2 and W24 are depicted with red lines. Statistical comparisons were performed using a Friedman test with Dunn's correction. **B&D** show fold-change in neutralizing titer against D614G **(B)** and BA.5 **(D)** between BL and W2 in each study arm. Bars represent median fold-change. A Kruskal-Wallis test with Dunn's correction was used to compare different arm groups. **(E)** Comparison of the neutralizing titer ($FRNT_{50}$) against D614G (left panel) and BA.5 (right panel) between study arms at W24. Bars represent GMT. Statistical comparisons were performed using a Kruskal-Wallis test with Dunn's correction.

In parallel, we assessed neutralizing antibody titers against an expanded panel of virus variants, namely D614G, Beta, Delta, Omicron BA.1 and BA.4/5 (which share identical spikes) using a lentiviral pseudovirus assay, limited to HIV-negative participants in the cohort. The data were consistent with our observations from the live virus assay, where significant boosting of neutralizing antibodies occurred at W2 against all variants for BNT162b2, but not for either the half- or full- dose Ad26.COV2.S (Fig 5). Titers against all variants declined from the W2 peak by W12, with a 2.4 to 5.7-fold drop, and levels remained constant to W24, with BNT162b2 arms trending to higher titers, regardless of the variant at W24.

Finally, we investigated antibody-dependent cellular cytotoxicity (ADCC) responses against the D614G, Beta, Delta and BA.1 spikes (Fig 6A and 6B) at BL and 2 weeks post-boost. As observed for the binding and neutralizing antibody responses, there was no increase in titers at W2 for either the half- or full-dose Ad26.COV2.S arms (median fold change 0.97 and 0.91, respectively), consistent with the fact that binding antibodies and ADCC potential are generally correlated. However, the half-dose BNT162b2 failed to trigger increased ADCC (median change 1.01), in contrast to the binding and neutralization results. For the full-dose BNT162b2 we observed marginally but nevertheless significantly higher ADCC titers against D614G, Beta and BA.1 spike at W2 compared to BL, but not to the same extent as binding or neutralizing responses.

## T-cell responses before and after booster vaccination

We also measured SARS-CoV-2 spike-specific T-cell responses before and after homologous and heterologous vaccination with full or half-dose vaccines (n = 214). Prior to boosting, spike-specific CD4+ responses were detected in most participants (>94.4%), with a frequency comparable between the four arms (Fig 7A). Two weeks after booster immunization, the frequency of spike-specific CD4+ T cells was significantly increased in participants who received BNT162b2 compared to BL (median: 0.23% vs 0.12% for half dose BNT162b2 and 0.25% vs 0.11% for full dose BNT162b2, respectively), while in participants boosted with Ad26.COV2.S, the median frequency of SARS-CoV-2 spike-specific CD4+ T cells, while statistically higher than BL, demonstrated only a marginal increase (0.12% vs 0.10% for half dose and 0.15% vs 0.11% for full dose) (Fig 7B). The median fold change in CD4+ response between BL and W2 was significantly greater after a BNT162b2 booster (1.8 for half dose and 1.98 for full dose) compared to an Ad26.COV2.S booster (1.1 for both half and full dose; Fig 7C). Overall, ~80% of participants boosted with BNT162b2 (regardless of the dose) had an increased spike-specific CD4+ T-cell response, while only 35.3% of participants who received half dose Ad26.COV2.S and 43.4% in those receiving a full dose Ad26.COV2.S booster expanded their CD4+ T-cell responses (Fig 7D). Spike-specific CD8+ T-cell responses were less frequent than CD4+ responses, detected in only ~55% of the participants at BL, and for those with a CD8+ response, the frequency of spike-specific CD8+ T cells was comparable between the four groups at BL (Fig 7E). Two weeks after boosting, a significant increase in the median CD8+ T-cell response was observed after a full-dose of Ad26.COV2.S (p = 0.002) and both half- or full-dose of BNT162b2 (p = 0.003 and p <0.001, respectively; Fig 7F). Assessing only participants with a spike CD8+ response at both time points, the median fold change at W2 post-boost was significantly higher after a BNT162b2 booster (1.91 for half-dose and 2.24 for full-dose) compared to an Ad26.COV2.S booster (1.22 for half-dose and 1.24 for full-dose; Fig 7G). However, it is important to note that individual responses were highly variable within all groups, with spike-specific CD8+ responses contracting, remaining negative or expanding. Overall, approximately half of participants boosted with BNT162b2 displayed an increase in spike-specific CD8+ T-cell responses, whereas an expansion of the CD8+ response was

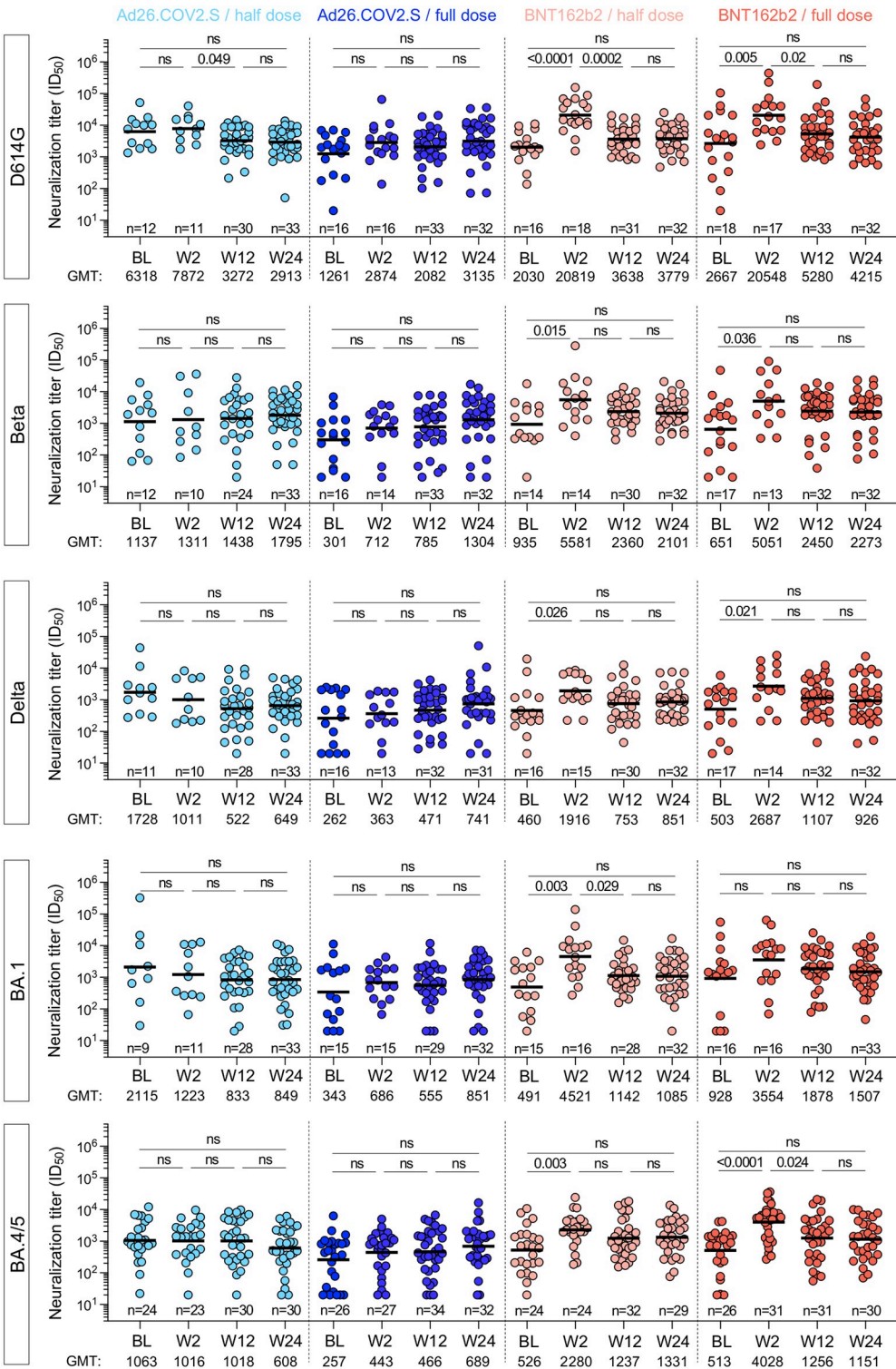

**Fig 5. Pseudovirus neutralization activity against ancestral D614G, Beta, Delta, BA.1 and BA.4/5 SARS-CoV-2 variants after booster vaccination.** Longitudinal neutralizing titer ($ID_{50}$) against ancestral D614G, Beta, Delta, Omicron BA.1 and BA.4/5 at BL, W2, W12 and W24 after vaccine booster. Bars represent medians. Bars represent GMT. Statistical comparisons were performed using a Kruskal-Wallis test with Dunn's corrections. Only HIV-negative participants were included in these analyses.

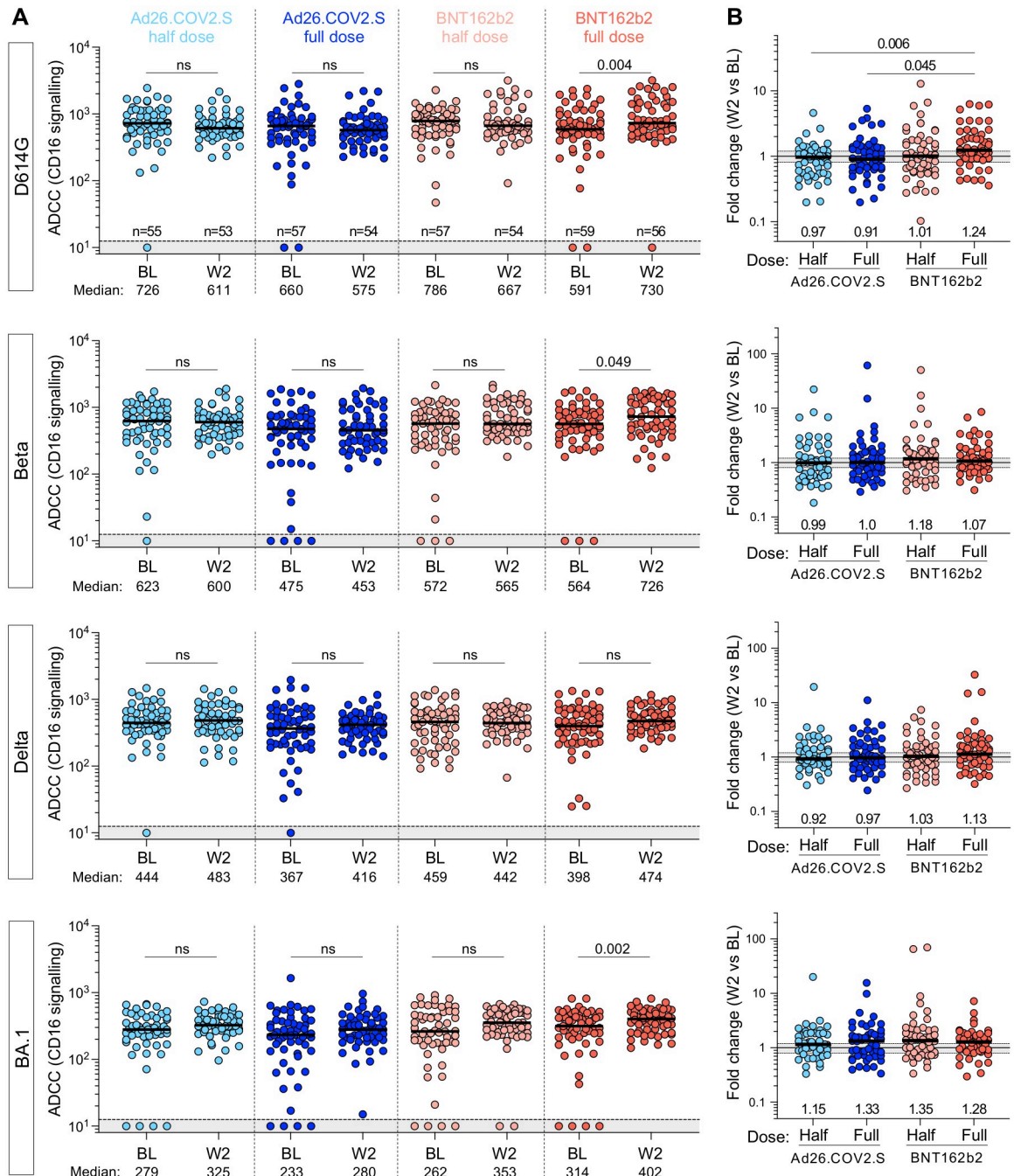

**Fig 6. Antibody-dependent cellular cytotoxicity (ADCC) against ancestral (D614G), Beta, Delta and Omicron BA.1 SARS-CoV-2 variants. (A)** ADCC (CD16 signalling) at BL and W2 in each study arm. Bars represent medians. The grey shaded area indicates an undetectable ADCC response. Statistical comparisons were performed using a Wilcoxon matched-pairs signed rank test. **(B)** Fold change in ADCC activity between W2 and BL in each study arm. Bars represent median fold change. Statistical comparisons were performed using a Kruskal-Wallis test with Dunn's corrections.

observed in only 1/3 of participants who received a half-dose Ad26.COV2.S booster and ~40% in those receiving a full-dose Ad26.COV2.S booster (Fig 7H).

The durability of T-cell responses was then assessed by measuring spike-specific T cells 24 weeks after the booster vaccination in a subset of participants (n = 190). Pairwise comparison

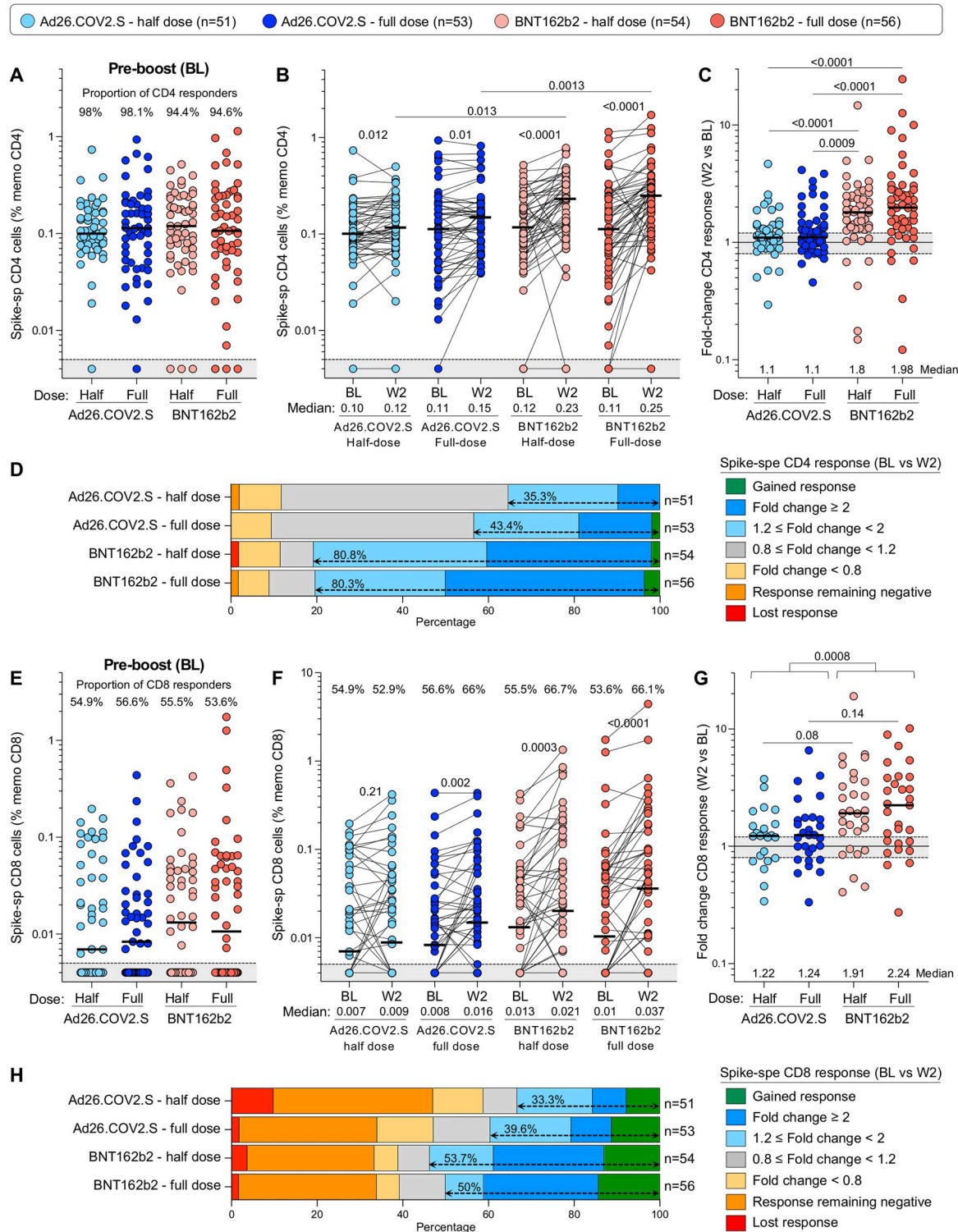

**Fig 7. SARS-CoV-2 spike-specific T-cell responses before and 2 weeks after vaccine boosting. (A)** Comparison of the frequency of spike-specific CD4+ T cells pre-boost in the four study arms. The grey shaded area indicates undetectable response. **(B)** Frequency of spike-specific CD4+ T cells before (BL) and after vaccine boost (W2). **(C)** Fold change in the frequency of spike-specific CD4+ T cells between W2 and BL. **(D)** Overall profile of the evolution of the spike-specific CD4+ T-cell response between BL and W2. **(E)** Comparison of the frequency of spike-specific CD8+ T cells pre-boost in the four trial arms. **(F)** Frequency of spike-specific CD8+ T cells before (BL) and after vaccine boost (W2). **(G)** Fold change in the frequency of spike-specific CD8+ T cells between W2 and BL. **(H)** Overall profile of

the evolution of the spike-specific CD8+ T-cell response between BL and W2. Bars represent medians. A two-tailed Wilcoxon signed-rank test was used to assess statistical differences between paired samples and a Kruskal-Wallis with Dunn's corrections was used to compare different groups.

of spike-specific CD4+ T-cell frequencies at W24 relative to W2 demonstrated a significant contraction in the CD4+ response for all groups (p <0.001 for both BNT162b2 boosters and half-dose Ad26.COV2.S and p = 0.0047 for full-dose Ad26.COV2.S), subsiding to BL levels (Fig 8A). Consequently, 24 weeks post-boosting, the magnitude of spike-specific CD4+ T cells was comparable in all trial arms (Fig 8B). In contrast, the median frequency of spike-specific CD8+ T cells appeared to increase between W2 and W24 in the Ad26.COV2.S group, although individual responses were highly variable, with some participants showing substantial waning or stable levels while others demonstrated gain of a CD8+ T-cell response that could be related to an undocumented BTI (Fig 8C). Overall, as for the CD4+ T-cell response, the frequency of spike-specific CD8+ T cells was similar across the four vaccine regimens when examined 24 weeks after booster vaccination (Fig 8D).

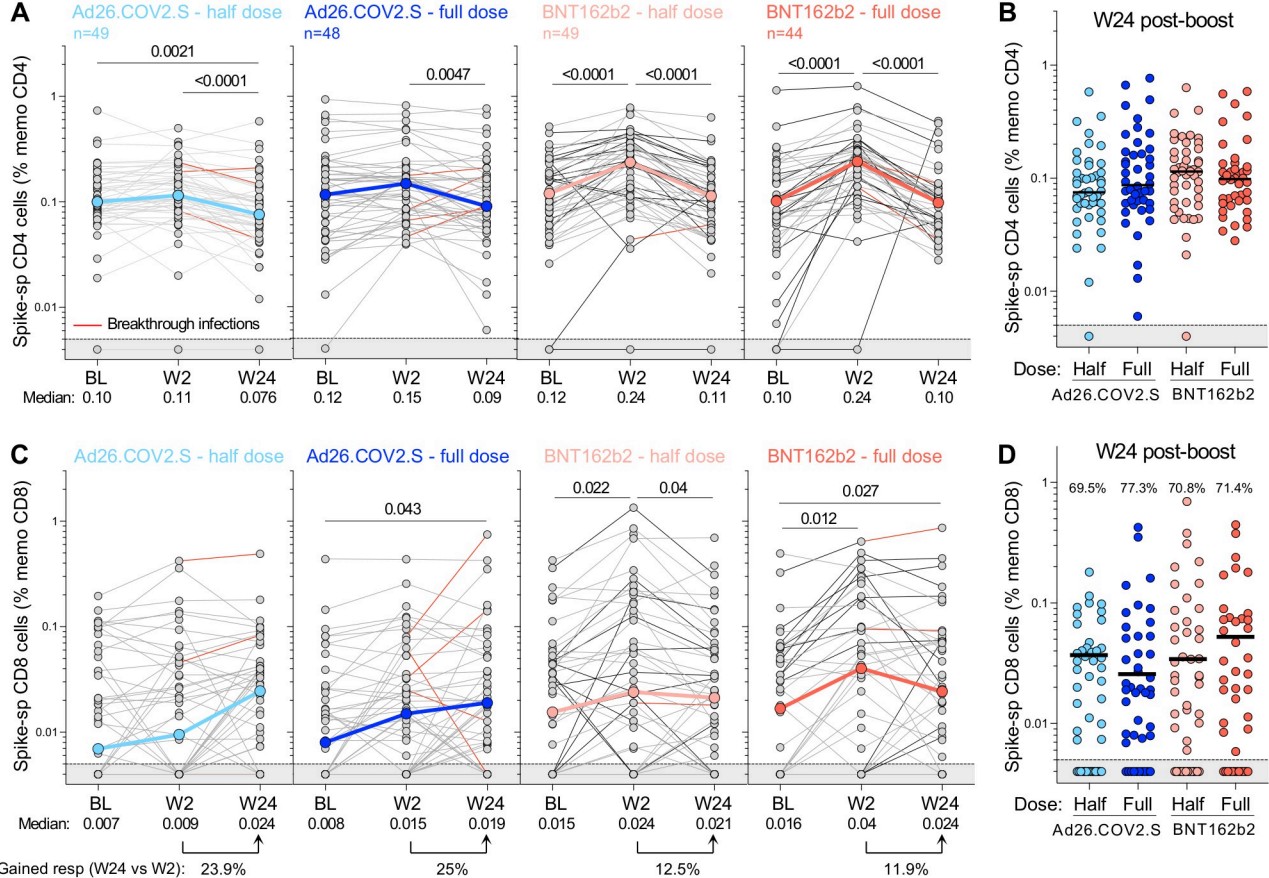

**Fig 8. Kinetics of SARS-CoV-2 spike-specific T-cell response after vaccination. (A)** Longitudinal frequencies of spike-specific CD4+ T-cell responses induced by the four different booster vaccine regimens. **(B)** Comparison of the frequency of spike-specific CD4+ T cells between the four arms at W24 post-boost. **(C)** Longitudinal spike-specific CD8+ T-cell responses induced by the four booster vaccine regimens. **(D)** Comparison of the frequency of spike-specific CD8+ T cells between the four arms at W24 post-boost. The proportion of spike CD8+ responders is indicated at the top of the graph. The grey shaded area indicates undetectable response. The color-coded dots and bold lines in (A) and (C) represent the median at each time point. Recorded BTI between W2 and W24 are depicted with a red line. A Friedman test with Dunn's correction was used to assess statistical differences between paired samples and a Kruskal-Wallis with Dunn's corrections was used to compare different groups.

## PLWH develop comparable antibody and T-cell responses after boosting

To determine whether HIV infection impacts vaccine booster responsiveness, participants were stratified by their HIV status. Prior to boosting, significantly lower titers of both binding and live virus neutralizing antibodies were found in PLWH who were viremic (viral load >200 HIV-1 mRNA copies/mL, n = 10) compared to virally suppressed individuals, who had similar antibody profiles to HIV-negative individuals (Fig 9A and 9B). Examination of baseline T-cell responses also revealed deficiencies, where significantly fewer viremic PLWH displayed spike-specific T-cell responses compared to aviremic and HIV-negative participants at BL (50% vs 100% and 97.6% for CD4+ and 10% vs 58.2% and 57.6% for CD8+; Fig 9C and 9D). Two weeks after booster vaccination, there was no difference in the degree of boosting (as measured by median fold-change between W2 and BL) for binding or neutralizing antibody titers between PLWH and HIV-negative participants for all vaccination groups (Fig 9E and 9F). For T-cell responses, however, the median fold change in CD4+ response in the full dose BNT162b2 group was significantly lower in PLWH compared to the HIV-negative participants (1.52 vs 2.21, respectively, p = 0.037), with a trend towards lower fold-change in CD8+ spike T-cell responses in PLWH (p = 0.065; Fig 9G and 9H). Of note, a high proportion of viremic participants did not mount any CD4+ (4/10) or CD8+ T-cell responses (4/10) after any of the vaccine boosts, indicative of the impact of immunosuppression (S1 Fig).

At W24, comparable titers of binding and neutralizing antibodies were detected between PLWH and HIV-negative participants in all vaccination groups, and T-cell frequencies did not show any deficiencies in PLWH (Fig 9I–9L). In fact, significantly higher frequencies of spike-specific CD4+ T cells were observed in PLWH in some vaccination groups (Fig 9K), possibly reflecting persistence of the higher baseline CD4+ T-cell responses observed in PLWH. Overall, these data suggest that some differences in the degree of boosting in PLWH, but by 24 weeks after revaccination, the frequencies of binding and neutralizing antibodies as well as spike-specific CD4+ and CD8+ T cells were comparable between HIV-negative participants and PLWH. This suggests that SARS-CoV-2 specific responses persist to a similar extent in those with well-controlled HIV infection compared to HIV-negative individuals. A proportion of viremic individuals, however, may be at risk of impaired T-cell responses and less durable antibody responses over the long term.

## Anti-vector immunity is detectable but may not account for the limited ability of Ad26.COV2.S to boost spike-specific responses

We hypothesized that the muted capacity of Ad26.COV2.S to boost spike responses, compared to BNT162b2, may be due to anti-vector immunity, given that prior studies demonstrated that pre-existing Ad5 vector-specific nAbs and T cells have the capacity to limit responses to the transgene immunogen [44–46]. To measure anti-Ad26 neutralizing antibody activity, we established an Ad26 neutralization assay in which dilutions of plasma from vaccinees were mixed with the Ad26.COV2.S vaccine, and vector infection was measured by spike expression in the Ad26 vector infected cells (Fig 10A and 10B). To quantify Ad26-specific neutralizing activity before and after boosting, we titrated participant plasma to obtain the plasma concentration needed to inhibit 50% of cellular infections with the Ad26 vector represented as $ID_{50}$, the reciprocal of this plasma dilution (Fig 10C). We tested 24 participants who had received full dose Ad26.COV2.S, and compared them to 24 participants who were boosted with BNT162b2, where no expansion of the vector-specific response would be expected. At BL, the majority of participants in both groups had Ad26-specific neutralizing antibody responses above the level of quantification (Fig 10D). Two weeks later, there was a significant increase in vector-specific $ID_{50}$ in the Ad26.COV2.S boosted group (p = 0.0006), but no boosting in the

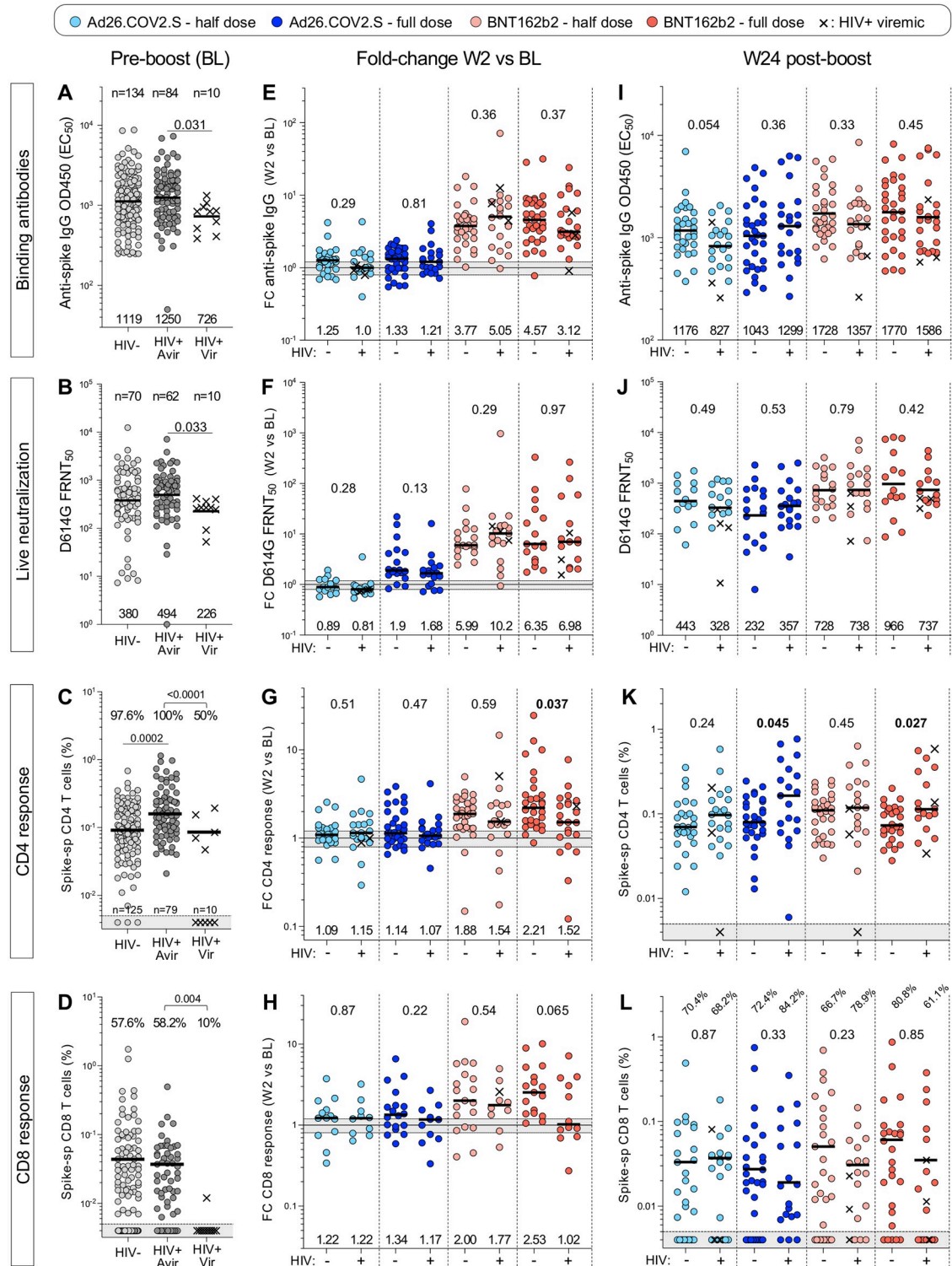

**Fig 9. Humoral and cellular responses in study participants stratified by HIV status. (A to D)** Spike-specific binding antibodies (A), live neutralization activity (B), spike-specific CD4+ response (C) and spike-specific CD8+ response (D) against ancestral (D614G) SARS-CoV-2 in HIV-negative (HIV-), PLWH with a viral load <200 HIV mRNA copies/ml (HIV+ Avir) and PLWH with a viral load >200 HIV mRNA copies/ml (HIV+ Vir) before vaccine booster (BL). Bars represent GMT for A, B, I, J and medians for all other graphs. Statistical comparisons were performed using a Kruskal-Wallis test with Dunn's corrections. The proportion of T-cell responders is indicated on top of the graph. **(E to H)** Fold change in spike-specific binding antibodies (E), live neutralization

activity (F), spike-specific CD4+ response (G) and spike-specific CD8+ response (H) between W2 and BL in each study arm, stratified by HIV status. **(I to L)** Spike-specific binding antibodies (I), live neutralization activity (J), spike-specific CD4+ response (K) and spike-specific CD8+ response (L) against ancestral (D614G) SARS-CoV-2 at W24 after vaccine booster. Viremic PLWH are identified with a cross. Bars represent medians. Statistical comparisons between PLWH and HIV-negative groups were performed using a Mann-Whitney test.

group receiving BNT162b2 (median fold-change in nAb $ID_{50}$ between W2 and BL: 1.83 for Ad26.COV2.S and 0.87 for BNT162b2; Fig 10E). To determine the level of neutralizing immunity against Ad26.COV2.S before primary Ad26.COV2.S vaccination, we investigated an independent cohort where participants were vaccinated with Ad26.COV2.S for the first time, with a subset accessing a second Ad26.COV2.S vaccine (S2 Table). The levels of anti-Ad26 neutralizing immunity prior to Ad26.COV2.S prime vaccination were below the level of quantification in all but one participant. Titers were significantly higher after an Ad26.COV2.S prime (p <0.0001; Fig 10F) but did not further increase after a boost, and were of similar magnitude to those in the BaSiS trial (Fig 10F). Finally, we investigated the association between Ad26-specific nAb titer at BL and the fold-change in spike neutralizing titer from W2 to BL, but found no correlation (Fig 10G).

To measure Ad26-specific T-cell responses, we stimulated PBMC with a pool of peptides spanning the viral hexon and penton proteins of Ad26 and characterized T-cell cytokine responses by flow cytometry. Ad26-specific T cells were readily detectable at BL and 2 weeks post-boosting (Fig 11A). We tested 44 participants who had received full-dose Ad26.COV2.S, and compared them to 20 participants who were boosted with BNT162b2, as a control group. At baseline, all participants in both groups had a CD4+ T-cell response to Ad26 (Fig 11B). Two weeks after Ad26.COV2.S booster vaccination, there was no overall increase in the Ad26-specific CD4+ response, which resembled the group who received the BNT162b2 booster. When comparing the fold-difference in the response from BL to W2, there was a minor increase in the Ad26.COV2.S-boosted group (1.08 for Ad26.COV2.S and 0.91 for BNT162b2), reflecting a proportion of participants who had a marginal increase in the frequency of Ad26-specific CD4+ T cells (Fig 11C). We also investigated an independent unvaccinated healthcare worker cohort prior to Ad26.COV2.S vaccination (S3 Table); cross-sectional analysis indicated a significant, 3-fold higher magnitude of Ad26-specific CD4+ T-cell responses 4 weeks after vaccination (p = 0.017; Fig 11D), similar to what we observed for Ad26 nAb responses. Interestingly, 85% of participants had an existing adenovirus CD4+ response even before the priming dose, which is the likely result of conserved T-cell cross-reactivity due to infection with other adenoviral types.

Similarly, we characterized CD8+ T-cell responses specific for Ad26. These were detectable in 75% of participants at baseline, and unlike CD4+ responses, were boosted significantly (p = 0.002) by a second dose of Ad26.COV2.S, demonstrating an increase in response of 1.2 compared to 0.95 for the BNT162b2-boosted group (Fig 11E and 11F). CD8+ responses were more rare in the cohort not previously vaccinated with Ad26.COV2.S, detectable in only 35% of unvaccinated participants. These were significantly higher (p = 0.005) and detectable in 81.8% of individuals 4 weeks after initial Ad26.COV2.S vaccination, at a magnitude and range similar to the baseline responses in our trial (Fig 11G). Importantly, and consistent with Ad26-specific nAb responses, there was no association between the magnitude of Ad26-specific CD4+ or CD8+ T cells and spike-specific T-cell responses, either at BL or W2 (Fig 11H). Overall, anti-vector humoral and cellular immunity was abundantly detectable in our trial participants, but neither appeared to have a clear effect on dampening spike-specific responses.

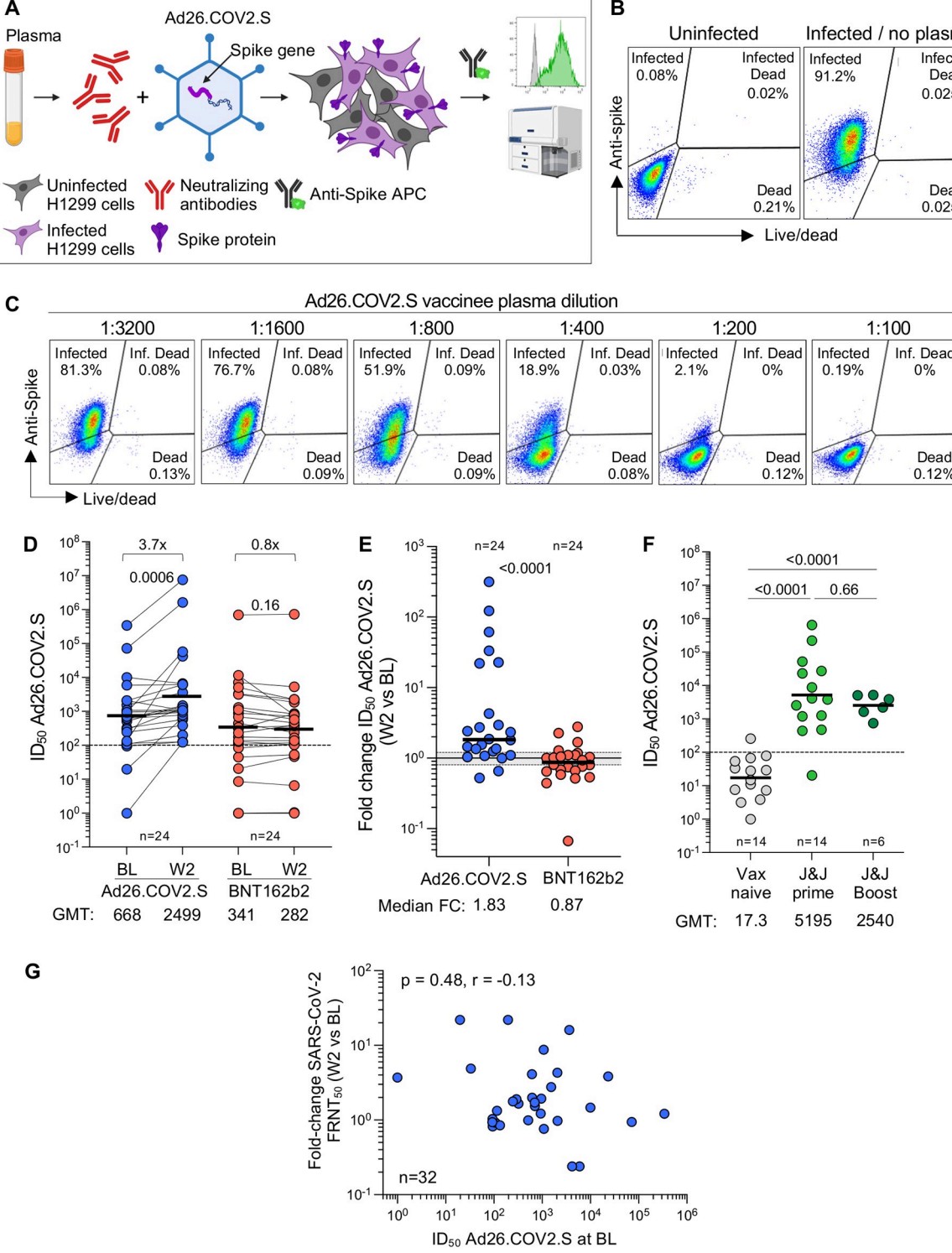

**Fig 10. Ad26-specific neutralizing activity. (A)** Schematic representation of the Ad26-specific neutralization assay. **(B)** Representative example of spike expression in Ad26.COV2.S-infected H1299 cells measured by flow cytometry. **(C)** Representative example of the inhibition of spike expression on Ad26.COV2.S-infected H1299 cells when Ad26.COV2.S was pre-incubated with plasma (serial dilution) from a participant vaccinated with one full dose of Ad26.COV2.S. **(D)** Ad26 neutralization activity ($IC_{50}$) pre- and W2 post full dose-Ad26. COV2.S or a full-dose BNT162b2 booster. Statistical difference were assessed using a Wilcoxon matched paired signed rank test. **(E)** Fold change in Ad26 neutralization activity between W2 and BL in Ad26.COV2.S or BNT162b2 boosted participants. Bars represent GMT for D

and F and medians for E. Statistical differences were assessed using a Wilcoxon matched paired signed rank test. **(F)** Comparison of Ad26 neutralization activity ($IC_{50}$) in individuals who were vaccine naïve (n = 14), received one full dose of Ad26.COV2.S (n = 14) or received two full doses of Ad26.COV2.S (n = 6) from an independent cohort. Statistical differences were assessed using a Kruskal-Wallis test with Dunn's correction. **(G)** Relationship between the fold change in neutralizing titer against D614G SARS-CoV-2 between W2 and BL and Ad26 neutralization activity at BL. Correlation was tested by a two-tailed non-parametric Spearman's rank test.

## Discussion

In this trial we evaluated the safety and immunogenicity of fractional and full doses of the Janssen adeno-vectored Ad26.COV2.S vaccine and the Pfizer BNT162b2 mRNA vaccine in an open label phase 2 trial of adults who had previously received a single dose of Ad26.COV2.S. The overwhelming majority of participants had detectable anti-nucleocapsid responses prior to receiving their boosts, indicating prior SARS-CoV-2 infection and boosting on the background of extensive hybrid immunity. We investigated safety and immunogenicity in both HIV-negative participants and PLWH, with the latter group generally well-controlled with VL suppression, and only 13.8% of PLWH being viremic (VL >200 cps/ml). Safety profiles in all four booster regimens were similar, with no regimen resulting in appreciable increased reactogenicity beyond localized responses at the injection site and mild, transient symptoms such as headache and weakness. Documented BTI, while few, mostly occurred in the Ad26.COV2.S boosted participants.

Heterologous boosting with the BNT162b2 mRNA vaccine was superior to homologous vaccination at early time points (2–12 weeks) by multiple humoral and cellular immune measures, including binding and neutralizing antibodies, and CD4+ T-cell responses to SARS-CoV-2 spike. This result is consistent with other studies showing better neutralizing antibody and T-cell responses after heterologous relative to homologous boosting of Ad26.COV2.S with an mRNA vaccine [23, 50–52]. Similar results were observed with mRNA vaccine heterologous boosting of the ChAdOx1 chimpanzee adeno-vectored vaccine [53–55]. Conversely, a homologous Ad26.COV2.S boost yielded mixed results. While some studies have demonstrated a significant increase (approximately 2- to 3-fold) in spike-specific IgG, neutralizing antibodies, and spike-specific T cell responses, irrespective of the dosage administered [56, 57], others have reported more modest effects, if any, in enhancing these immune responses [23, 50–52]. These differences could be related to several factors, including an individual's previous history of SARS-CoV-2 infection, the interval between doses, the timing between doses and/or the potential impact of pre-existing or vaccine-induced anti-vector immunity.

The superiority of heterologous vaccination over homologous vaccination is reinforced by a vaccine efficacy study showing that vaccine efficacy was highest for regimens incorporating a booster dose of an mRNA vaccine and lowest for the Ad26.COV2.S/Ad26.COV2.S regimen [58]. This could be related to the ability of heterologous vaccination (Ad26.COV2.S followed by BNT162b2) to broaden the humoral and cellular spike-specific repertoire, thereby expanding antibodies and T cells capable of recognizing the S2 region of Spike [50].

In our study, binding antibodies, neutralization, and T-cell responses increased 2-fold or less, and halving the dose resulted in no detectable booster activity. In contrast, boosting with BNT162b2 gave a 4- to 10-fold increase of antibody responses, with the exception of ADCC (discussed below), and 80% of trial participants had increased spike specific CD4+ T-cell responses. Unlike with Ad26.COV2.S, halving the dose of BNT162b2 did not strongly decrease the response to the booster in terms of either elicited binding antibodies, neutralizing antibodies, or CD4+ T-cell responses. Despite the strong boosting effect of BNT162b2, waning of the binding and neutralizing antibody responses was pronounced by 12 weeks post-boost and by week 24, titers had declined close to the relatively high baseline titers. For ADCC, which

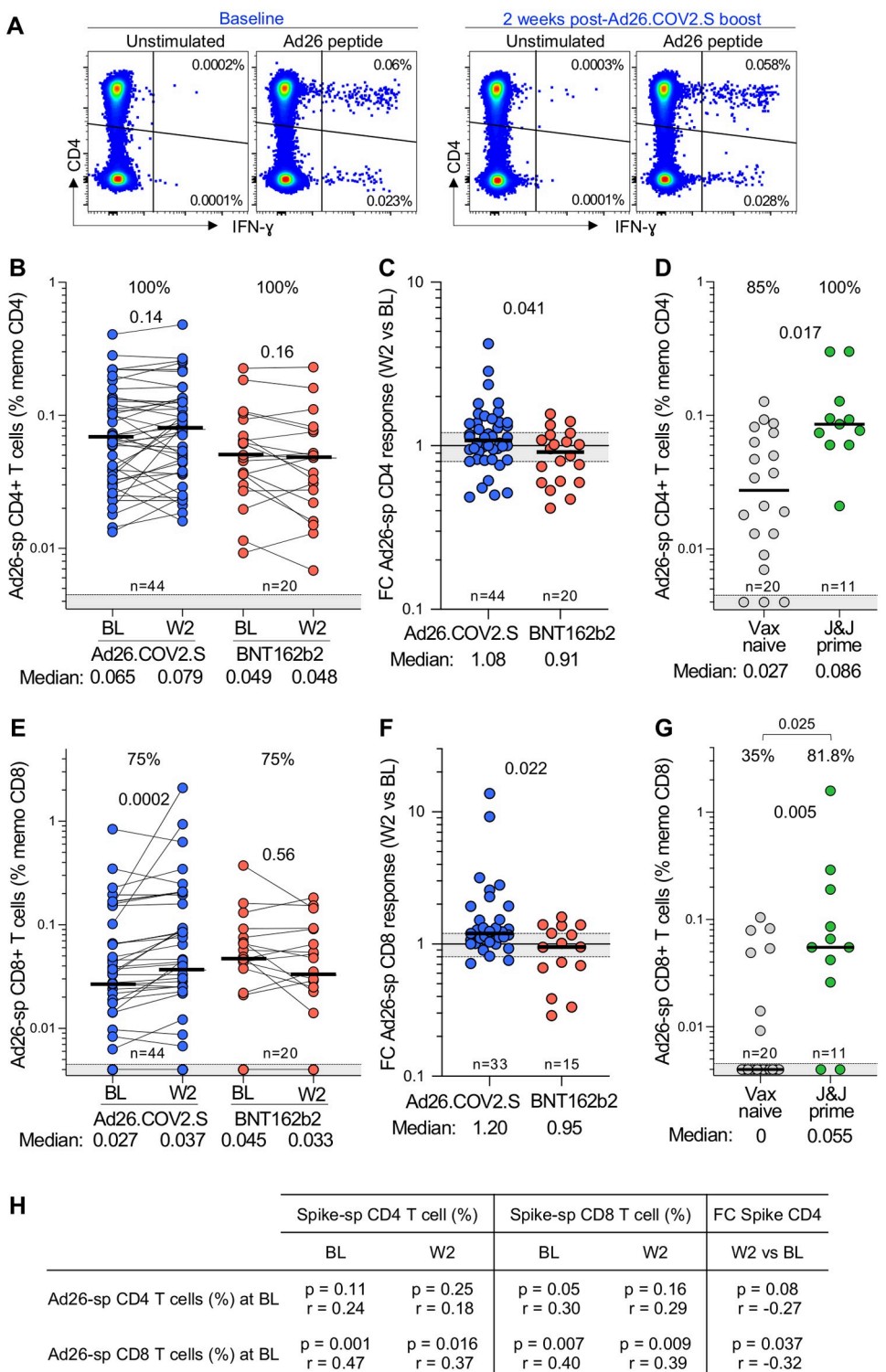

**Fig 11. Ad26-specific T-cell responses. (A)** Representative example of IFN-g production in response to Ad26-specific peptide pool (hexon and penton) in one participant before (baseline) and 2 weeks after a full dose-Ad26.COV2.S booster. **(B & E)** Frequency of Ad26-specific CD4+ T cell (B) and CD8+ T cells (E) pre- and post a full dose-Ad26. COV2.S or a full-dose BNT162b2 booster. Statistical difference were assessed using a Wilcoxon matched paired signed rank test. **(C & F)** Fold change in the frequency of Ad26-specific CD4+ T cells (C) and CD8+ T cells (F) between W2 and BL in full-dose Ad26.COV2.S or full-dose BNT162b2 boosted participants. **(D & G)** Comparison of the frequency

of Ad26-specific CD4+ T cells (D) and CD8+ T cells (G) in individuals who are vaccine naïve (n = 20) or received one full dose of Ad26.COV2.S (n = 11) from an independent cohort. The proportion of Ad26 T- cell responders is indicated at the top of each graph. Statistical differences were assessed using a Kruskal-Wallis test with Dunn's correction and a Chi-test to compare proportions. Bars represent medians. (H) Relationship between the frequency of spike-specific T-cell response at BL or W2 and the frequency of Ad26-specific T-cell responses at BL. Correlations were tested by a two-tailed non-parametric Spearman's rank test.

generally correlates well with binding and neutralization, the absence of boosting by Ad26. COV2.S is expected, given the lack of boosting overall. However the marginal ADCC boosting observed only in the full-dose BNT162b2 arms (despite increased binding and neutralization) may be a consequence of class switching towards the IgG4 subclasses, which does not effectively mediate ADCC, as recently reported for repeated mRNA vaccination [59, 60].

The effects of HIV status on the neutralizing antibody and T-cell responses were moderate, and largely limited to reduced responses observed at baseline in PLWH with HIV viremia (despite the small size of this group). Both binding neutralizing antibody responses to ancestral SARS-CoV-2 were significantly lower in HIV viremic participants at baseline, while there was no difference between aviremic PLWH and HIV-negative participants. CD4+ T-cell responses showed a more marked difference with only 50% of viremic PLWH having detectable spike-specific CD4+ T cells, compared with 100% of aviremic PLWH and 97% of HIV-negative participants. For CD8+ T cells, the fraction of detectable spike-specific cells was 58% for both aviremic PLWH and HIV negative participants. In contrast, only 1 out of 10 HIV viremic participants had a detectable spike-specific CD8+ T-cell response. Interestingly, the median fraction of spike specific CD4+ T cells was significantly higher in aviremic PLWH than the HIV-uninfected participants, perhaps indicative of a different SARS-CoV-2 course of infection in PLWH [27] or interactions of SARS-CoV-2 immunity with HIV-mediated partial immune activation in HIV suppressed PLWH [61].

Post-boost, binding antibodies and neutralization were generally similar between PLWH and HIV negative participants. Even viremic PLWH showed increased binding and neutralizing antibody responses after BNT162b2 vaccination, and these were within the range seen in the other participants. These results are consistent with previous studies that PLWH have good antibody responses to vaccination [19, 38, 39, 62] with the exception of those with CD4+ T cell concentrations than 200 cells/mm$^3$ [34, 36, 37, 63, 64]. T-cell responses were more strongly affected by HIV status: they were significantly lower for spike-specific CD4+ T cells in PLWH boosted with a full-dose of BNT162b2. We did not detect an increase in spike-specific CD8+ T-cell responses with full-dose BNT162b2, although the results did not reach statistical significance. The interpretation of the CD4+ T cell results is complicated by the higher median absolute fraction of spike-specific CD4+ T cells in PLWH post- BNT162b2 boost.

Although the Ad26.COV2.S is known to trigger lower responses than BNT162b2, the limited effect observed after homologous boosting in our cohort was striking. We thus assessed whether this was caused by anti-vector immunity. Vaccination with adenovirus 5 vectored vaccines elicits both neutralizing antibody and T-cell immunity to the vector itself, decreasing the efficiency of cellular infection with the vaccine vector and attenuating the vaccine response [44–46]. Using the samples from this trial and supplementing with other cohorts where a pre-Ad26.COV2.S prime vaccination sample was available, we observed negligible Ad26.COV2.S neutralizing activity in vaccine-naïve donors. However, a single dose of Ad26.COV2.S strongly increases both neutralizing titers against Ad26.COV2.S and the fraction of Ad26-specific CD4+ and CD8+ T cells. Nevertheless, as with other studies of anti-Ad26.COV2.S

neutralization, we could not find a clear association between the levels of anti-vector immunity, either neutralizing or T cell, and the degree of immune response post-Ad26.COV2.S boost, as previously reported [47, 48].

To conclude, we show that in the context of high levels of hybrid immunity, heterologous boosting with the BNT162b2 mRNA vaccine following a Ad26.COV2.S prime demonstrated superior immunogenicity and was safe and effective in both PLWH and HIV-negative participants, although there was a rapid waning of binding and neutralizing antibody responses. The Ad26.COV2.S homologous boost was also safe but showed highly attenuated immunogenicity relative to BNT162b2, both in antibody and T-cell immunity. While the Ad26.COV2.S vaccine elicited strong anti-vector immunity, we did not find clear evidence that this was the cause of attenuation.

## Supporting information

**S1 Protocol. Phase II randomised open label trial of full and half dose J&J Ad26.CoV2.S and Pfizer BNT162b2 booster vaccinations after receiving the J&J Ad26.CoV2.S prime vaccine through the SISONKE phase IIIB implementation study or through the South African COVID-19 vaccination programme.**
(PDF)

**S1 Checklist. Reporting checklist for randomised trial.**
(PDF)

**S1 Fig. SARS-CoV-2 spike-specific T cell responses in participants stratified by HIV status in each study arm.**
(DOCX)

**S1 Table. Dataset for Figs 2 to 11.**
(XLSX)

**S2 Table. Clinical characteristics of samples used for Ad26-specific antibody response assessment.**
(DOCX)

**S3 Table. Clinical characteristics of samples used for Ad26-specific T cell response assessment.**
(DOCX)

## Acknowledgments

We thank all the trial participants who contributed to this study.

## Author Contributions

**Conceptualization:** Frances Ayres, Helen Rees, Linda-Gail Bekker, Glenda Gray, Wendy A. Burgers, Alex Sigal, Penny L. Moore, Lee Fairlie.

**Data curation:** Jinal N. Bhiman, Katherine Gill, Erica Lazarus, Jean Le Roux, Gila Lustig, Anusha Nana, Ravindre Panchia, Faeezah Patel, Upasna Singh, Nigel Garrett, Lee Fairlie.

**Formal analysis:** Catherine Riou, Jinal N. Bhiman, Yashica Ganga, Simone I. Richardson, Wendy A. Burgers, Alex Sigal, Penny L. Moore, Lee Fairlie.

**Funding acquisition:** Linda-Gail Bekker, Glenda Gray.

**Investigation:** Catherine Riou, Jinal N. Bhiman, Yashica Ganga, Shobna Sawry, Frances Ayres, Gila Lustig, Nigel Garrett, Helen Rees, Wendy A. Burgers, Alex Sigal, Penny L. Moore, Lee Fairlie.

**Methodology:** Catherine Riou, Jinal N. Bhiman, Yashica Ganga, Shobna Sawry, Richard Baguma, Sashkia R. Balla, Ntombi Benede, Mallory Bernstein, Asiphe S. Besethi, Sandile Cele, Carol Crowther, Mrinmayee Dhar, Sohair Geyer, Tandile Hermanus, Haajira Kaldine, Roanne S. Keeton, Prudence Kgagudi, Khadija Khan, Mashudu Madzivhandila, Siyabulela F. J. Magugu, Zanele Makhado, Nelia P. Manamela, Qiniso Mkhize, Paballo Mosala, Thopisang P. Motlou, Hygon Mutavhatsindi, Nonkululeko B. Mzindle, Rofhiwa Nesamari, Amkele Ngomti, Anathi A. Nkayi, Thandeka P. Nkosi, Millicent A. Omondi, Strauss van Graan, Elizabeth M. Venter, Avril Walters, Thandeka Moyo-Gwete, Simone I. Richardson, Nigel Garrett, Wendy A. Burgers, Penny L. Moore.

**Project administration:** Shobna Sawry, Erica Lazarus, Jean Le Roux, Anusha Nana, Ravindre Panchia, Faeezah Patel, Upasna Singh, Glenda Gray.

**Resources:** Alba Grifoni, Alessandro Sette.

**Supervision:** Catherine Riou, Nigel Garrett, Wendy A. Burgers, Alex Sigal, Penny L. Moore.

**Validation:** Yashica Ganga, Lee Fairlie.

**Visualization:** Simone I. Richardson.

**Writing – original draft:** Catherine Riou, Jinal N. Bhiman, Shobna Sawry, Wendy A. Burgers, Alex Sigal, Penny L. Moore, Lee Fairlie.

**Writing – review & editing:** Catherine Riou, Jinal N. Bhiman, Yashica Ganga, Shobna Sawry, Frances Ayres, Richard Baguma, Sashkia R. Balla, Ntombi Benede, Mallory Bernstein, Asiphe S. Besethi, Sandile Cele, Carol Crowther, Mrinmayee Dhar, Sohair Geyer, Katherine Gill, Alba Grifoni, Tandile Hermanus, Haajira Kaldine, Roanne S. Keeton, Prudence Kgagudi, Khadija Khan, Erica Lazarus, Jean Le Roux, Gila Lustig, Mashudu Madzivhandila, Siyabulela F. J. Magugu, Zanele Makhado, Nelia P. Manamela, Qiniso Mkhize, Paballo Mosala, Thopisang P. Motlou, Anusha Nana, Rofhiwa Nesamari, Amkele Ngomti, Anathi A. Nkayi, Thandeka P. Nkosi, Millicent A. Omondi, Ravindre Panchia, Faeezah Patel, Upasna Singh, Strauss van Graan, Avril Walters, Thandeka Moyo-Gwete, Simone I. Richardson, Nigel Garrett, Helen Rees, Linda-Gail Bekker, Glenda Gray, Alex Sigal, Penny L. Moore, Lee Fairlie.

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
