## [Decision Letter · Decision Letter 0]

24 Jan 2024

PGPH-D-23-02273

Safety and immunogenicity of booster vaccination and fractional dosing with Ad26.COV2.S or BNT162b2 in Ad26.COV2.S-vaccinated participants

Dear Dr. Riou,

Thank you for submitting your manuscript to PLOS Global Public Health. Firstly, we would like to apologize for the delay in processing your manuscript. It has been exceptionally difficult to secure reviewers to evaluate your study. We have now received one completed review, which is available below. The reviewer has raised significant scientific concerns about the study that need to be addressed in a revision.

Please note that we have only been able to secure a single reviewer to assess your manuscript. We are issuing a decision on your manuscript at this point to prevent further delays in the evaluation of your manuscript. Please be aware that the editor who handles your revised manuscript might find it necessary to invite additional reviewers to assess this work once the revised manuscript is submitted. However, we will aim to proceed on the basis of this single review if possible.

We look forward to receiving your revised manuscript.

Kind regards,

Miquel Vall-llosera Camps

Staff Editor

Journal Requirements:

Additional Editor Comments (if provided):

Reviewers' comments:

Reviewer's Responses to Questions

**Comments to the Author**

1. Does this manuscript meet PLOS Global Public Health’s publication criteria? Is the manuscript technically sound, and do the data support the conclusions? The manuscript must describe methodologically and ethically rigorous research with conclusions that are appropriately drawn based on the data presented.

Reviewer #1: Yes

2. Has the statistical analysis been performed appropriately and rigorously?

Reviewer #1: Yes

3. Have the authors made all data underlying the findings in their manuscript fully available (please refer to the Data Availability Statement at the start of the manuscript PDF file)?

Reviewer #1: Yes

4. Is the manuscript presented in an intelligible fashion and written in standard English?

Reviewer #1: Yes

5. Review Comments to the Author

Reviewer #1: Riou et al in the manuscript “Safety and immunogenicity of booster vaccination and fractional dosing with Ad26.COV2.S or BNT162b2 in Ad26.COV2.S-vaccinated participants” investigated the antibody and T-cell responses upon either a full or half-dose booster of Ad26.COV2.S or BNT162b2 vaccine. They noted an increased immunogenicity after heterologous immunization compared with the homologous regimen. This study is rationally designed and the provided data justify the overall conclusion.

I agree with what the authors stated “…the almost complete absence of boosting by this vaccine (Ad26.Cov2.S) in our cohort was striking” (page 25, lines 638-639). Moreover, I think the Ad26 booster data overall is in discordance with the previous studies in humans (e.g., Interim Results of a Phase 1–2a Trial of Ad26.COV2.S Covid-19 Vaccine, Sadoff et al. NEJM 2021, and Immunogenicity of the Ad26.COV2.S Vaccine for COVID-19, Stephenson et al. JAMA 2021) and in macaques (A homologous or variant booster vaccine after Ad26.COV2.S immunization enhances SARS-CoV-2–specific immune responses in rhesus macaques, He et al. Sci Trans Med, 2022). Can the authors provide comments on that?

6. PLOS authors have the option to publish the peer review history of their article (what does this mean?). If published, this will include your full peer review and any attached files.

**Do you want your identity to be public for this peer review?** For information about this choice, including consent withdrawal, please see our Privacy Policy.

Reviewer #1: No

---

## [Decision Letter · Decision Letter 1]

20 Feb 2024

Safety and immunogenicity of booster vaccination and fractional dosing with Ad26.COV2.S or BNT162b2 in Ad26.COV2.S-vaccinated participants

PGPH-D-23-02273R1

Dear Dr Riou,

We are pleased to inform you that your manuscript 'Safety and immunogenicity of booster vaccination and fractional dosing with Ad26.COV2.S or BNT162b2 in Ad26.COV2.S-vaccinated participants' has been provisionally accepted for publication in PLOS Global Public Health.

Best regards,

Jerome Nyhalah Dinga, PhD

Academic Editor
